



# WoSIS: Serving standardised soil profile data for the world

Niels Batjes[1], Eloi Ribeiro[1], Ad van Oostrum[1], Johan Leenaars[1], Tom Hengl[1], and Jorge Mendes de Jesus[1]

[1]ISRIC – World Soil Information, Wageningen, 6708 PB, The Netherlands

*Correspondence to*: Niels H. Batjes (niels.batjes@isric.org)

**Abstract.** The aim of the World Soil Information Service (WoSIS) is to serve quality-assessed, georeferenced soil data (point, polygon, and grid) to the international community upon their standardisation and harmonisation. So far, the focus has been on developing procedures for legacy point data, with special attention for the selection of soil analytical and physical properties considered in the *GlobalSoilMap* specifications (e.g. organic carbon, soil pH, soil texture (sand, silt, and clay), coarse fragments (< 2mm), cation exchange capacity, electrical conductivity, bulk density, and water holding capacity).

Profile data managed in WoSIS were contributed by a wide range of soil data providers; the data have been described, sampled, and analysed according to methods and standards in use in the originating countries. Hence, special attention was paid to measures for soil data quality and the standardisation of soil property definitions, soil property values, and soil analytical method descriptions. The present version of WoSIS contains some 118,400 unique 'shared' soil profiles of which over 96,000 are georeferenced within defined limits. In total, this corresponds with over 31 million soil records, of which

some 20% have so far been quality-assessed and standardized using the procedures discussed in this paper. The number of measured data for each property varies between profiles and with depth, generally depending on the purpose of the initial studies. Overall, the data lineage strongly determined which data could be standardized with acceptable confidence (as flagged in the database). The publically available data –WoSIS snapshot of July 2016– are persistently accessible from ISRIC WDC-Soils through http:\\dx.doi.org\10.17027\isric-wdcsoils.20160003.

## 1 Introduction

Soil is an important provider of ecosystem services (UNEP, 2012; MEA, 2005). Yet this natural resource, considered to be non-renewable on a human life span, is being threatened (FAO and ITPS, 2015; UNEP, 2014). Worldwide, professionals, scientists, "decision makers and managers must have access to the information they need, when they need it, and in a format

they can use" (GEO, 2010). Large numbers of consistent soil profile data of known provenance (lineage) are needed to accurately model and map the status of the world's soil resources at increasingly detailed resolutions (Leenaars et al., 2014; Omuto et al., 2012; Hengl et al., 2014; Arrouays et al., 2014; FAO and ITPS, 2015).

This paper describes procedures for safeguarding, standardising/harmonising and subsequently serving of consistent world soil data to the international community as developed in the framework of the Data\WoSIS (World Soil Information Service)





project. In essence, the development of the WoSIS server database may be seen as a sequel to earlier collaborative, but still 'stand-alone', compilations of soil legacy data coordinated by ISRIC such as WISE (Batjes, 2009), SOTER (van Engelen and Dijkshoorn, 2013), and the Africa Soil Profiles database (Leenaars et al., 2014). Ultimately, WoSIS aims to serve consistent harmonized soil data (point, polygon, grids), derived both from a wide range of shared legacy holdings as well as

5    from recently developed soil spectral libraries (e.g. Viscarra Rossel et al., 2016; Shepherd and Walsh, 2002), in an interoperable mode and this preferably in the setting of a federated global soil information system.

Harmonisation, as defined by the Global Soil Partnership (GSP, Baritz et al., 2014), involves 'providing mechanisms for the collation, analysis and exchange of consistent and comparable global soil data and information'. Areas of harmonisation include those related to: a) soil description, classification and mapping, b) soil analyses, c) exchange of digital soil data, and

10   d) interpretations. So far, seen the breadth and magnitude of the task, the focus in WoSIS has been on the standardisation of soil property definitions, soil analytical method descriptions and soil property values, for those properties considered in the GlobalSoilMap specifications (GlobalSoilMap, 2013). Such a standardisation is a prerequisite for the development/testing of a soil information model that can underpin global soil data interoperability and modelling (Omuto et al., 2012). Quality assessed profile data served from WoSIS, and its predecessors as discussed above, may be used for various purposes such as

conventional resp. digital mapping of soil properties and soil classes (Batjes, 2016; Hengl et al., 2015; Hengl et al., submitted, 2016); in turn, such derived products may be used in studies that address a range of global issues at various scale levels  (e.g. Hendriks et al., 2016; Luo et al., 2016; Jones and Thornton, 2015; Maire et al., 2015).

## 2 Data and methods

### 2.1 Basic principles

Everyone may contribute data for inclusion in WoSIS. The access rights and data provenance (lineage), as documented in the metadata, will determine which quality assessed-data may later be served freely to the international community. Therefore, when processing the wealth of contributed data, priority is given to those datasets that have a 'non-restrictive' Creative Commons licence, defined here as at least a CC BY (Attribution) or CC BY-NC (Attribution Non-Commercial). The

corresponding source data will be gradually standardised and harmonised to make them 'comparable as if assessed by a given (reference) method' (Figure 1). Ultimately, only the quality-assessed and standardised/harmonised 'shared' data will be served to the international community.

<< Insert Figure 1 >>



## 2.2 Measures for data quality

As indicated, soil profile data submitted for consideration in WoSIS were collated according to various national or international standards. Therefore, proper documentation, in so far as possible, of the provenance and identification of each dataset, and ideally each observation or measurement, is essential to allow for efficient processing of the data; such aspects

are discussed later in detail.

For soil observations and measurements, the following need to be specified: feature (x-y-z and time (t) referenced profiles and layers), attribute (class, site, layer-field, and layer-lab), method, and value, including units of expression (Leenaars, 2012; Leenaars et al., 2014; Ribeiro et al., 2015). As indicated by Chapman (2005), "too often, data are used uncritically without consideration of the error contained within, and this can lead to erroneous results, misleading information, unwise

environmental decisions and increased costs". WoSIS is being populated using data produced for different types of studies ranging from routine soil surveys to more specific assessments, each of these having their specific quality requirements (Landon, 1991; Soil Survey Division Staff, 1993). The corresponding samples were analysed in a range of laboratories or in the field according to a wide range of methods (e.g. wet chemistry or soil spectroscopy), each with their own uncertainty and costs. As indicated by Kroll (2008), issues of soil data quality are not restricted to uncertainty issues; they also include

aspects like completeness, accessibility and verifiability (traceability) of data.

A review of quality aspects specifically related to soil data led to consideration of four quality indicators in WoSIS (Ribeiro et al., 2015): (a) observation date (date of observation or measurement), (b) level of trust (a subjective measure inferred from soil expert knowledge), (c) data quality rating (based on expert judgement), and (d) accuracy (an indicator for the laboratory and field-related uncertainty as well as the accuracy of georeferencing). These indicators provide measures that allow

investigators to recognize factors that may compromise the quality of certain data and hence their suitability for use. Consideration of such quality indicators ensures that objective methods are applied for evaluating data in the database, while at the same time the system enables soil expert knowledge to override these assessments when needed (Ribeiro et al., 2015). In practice, however, the information provided with some source materials does not allow for a full characterization of all four indicators.

## 2.3 Standard data model

Sometimes, the source data may be in paper (analogue) format in which case they must first be digitized following certain basic criteria (Ribeiro et al., 2015, p. 37-40; Leenaars et al., 2014, p. 52). Preferably, data entry is done by the data providers, as they best know their data. Basically, this first step amounts to 'safeguarding soil data collections' at risk of being lost, which is an important remit of ISRIC as World Data Centre for Soils (WDC-Soils) of the ICSU World Data System.

Any submitted digital data set is first assessed as regards its overall suitability for inclusion in WoSIS (e.g. licence and metadata). After this filtering, the data are imported 'as is' into PostgresSQL, an open source database management system. At this early stage of processing, the data are still organised according to numerous data models, data conventions and data



contents. So, the next step (yet first step of standardisation) is to map this disparate data to the WoSIS *standard data* model (Figure 1); technical details are provided elsewhere (Ribeiro et al., 2015).

## 2.4 Identifying repeated profiles

Being derived from multiple data sources, some of which are compilations, there is a risk that the same profiles are imported several times into WoSIS *albeit* using different identifiers. Computerised procedures that consider lineage and geographical proximity checks were developed to screen for possible repeated profiles. Figure 2 serves to show the results of this time-consuming process for four databases: ISIS (2016), the ISRIC Soil Information System (reference collection); WISE, World Inventory of Soil Emission potentials (compilation, Batjes, 2009); SOTER, Soil and Terrain databases (compilation, Van Engelen, 2011); and AfSP, the Africa Soil Profiles database (compilation, Leenaars et al., 2014). For example, 12,810 profiles are present only in AfSP, 35 are shared among AfSP and ISIS (the original source), 164 are shared between AfSP, WISE and ISIS, and 10 profiles occur in the four databases. In case of duplicate profiles, only the profile with the most complete data and detailed lineage is maintained for further processing.

<< Insert Figure  2 >>

## 2.5 Basic data quality assessment and control

For all source data –subsequent to some basic QA/QC checks– it is assumed that the quality requirements of the (first) user are met and that basic quality checks and screening have taken place, this with due consideration for any soil-specific options in the laboratory procedures (Ribeiro et al., 2015). This approach allows users of WoSIS to make their own judgement on the quality of individual analytical data, for instance by the assumption that selected data have comparable quality characteristics or an acceptable (inferred) quality compared to their requirements.

## 2.6 Standardisation of soil analytical method descriptions

As indicated, there is often no detailed quantitative information on the quality and uniformity of the soil analytical data held in the diverse source databases. Full quality control, including verification of in-pedon consistencies, requires the data to be harmonised according to an analytical reference method. The foreseen ultimate step of data harmonisation, converting property values assessed with analytical method X to values 'as if' assessed by reference method Y, requires an unambiguous identification and definition of the various analytical methods. Therefore, it was necessary to develop a



qualitative procedure to describe the analytical methods, including their method features, in a flexible, yet comprehensive and consistent way.

The options selected for the analytical method features in WoSIS are assigned on basis of the descriptions in the respective (database) sources. This implies that information, as interpreted or distilled from the original report (source materials) by the data compilers, was used in WoSIS. In the future, some refinements may still prove possible or necessary should the original materials, such as laboratory manuals, be consulted again.

In essence, the WoSIS approach for the qualitative description of soil analytical methods can be seen as complementary to method descriptions used in reports from proficiency tests (NATP, 2015; van Reeuwijk, 1998; WEPAL, 2015). In such tests, results from participants are coded to provide details of the methods applied for a particular grouping (e.g. CEC, cation exchange capacity). As discussed in Ribeiro et al (2015), the spread of these results may give an indication for the maximum spread in a compiled database.

In addition to the method description according to the standardised coding system developed for WoSIS, measures have been allocated for the inferred confidence in each 'method conversion' (i.e. from low to high); of necessity, this qualitative assessment is based solely on the information embedded in the 'summarised' method descriptions as provided in the various source databases. As indicated, such descriptions have often been generalised from a more detailed source, such as a laboratory manual. Importantly, the provided confidence flags should not be seen as a measure for the quality of a particular laboratory (see Ribeiro et al., 2015).

**2.7 Towards the harmonisation of world soil data**

Depending on the projected applications, user communities will require specific sets of data. In first instance, we limited ourselves to the list of properties considered in the GlobalSoilMap specifications (GlobalSoilMap, 2013): soil pH, soil organic matter content, effective cation exchange capacity, electrical conductivity, soil texture (sand, silt, and clay content), proportion of fragments > 2 mm, bulk density and water retention. In the respective source databases, these properties were determined using a range of analytical procedures, thus requiring standardisation of the soil analytical method descriptions to make them 'fit for use' and comparable (Leenaars et al., 2014). Key in the approach developed for WoSIS is that 'a property is best described by key elements of the (laboratory) procedure applied' (Soil Survey Staff, 2011). Similarly, in WoSIS, major features of commonly used methods for determining a given soil property are characterised. For soil pH, for example, these are the solution, concentration, ratio (soil/solution), and instrument. As indicated, the key component features can be aggregated where considered as being comparable in the context of global or regional level data analyses. For example, soil pH data measured in a KCl solution, 1M, at a soil/liquid ratio of 1:5, and using a conventional electrode can be aggregated within the group considered to meet the ISO 10390:2005 criteria for pH-KCl (ISO, 2015). Similarly, the combination KCl solution, 1 M, 1:2.5 soil/liquid ratio, and conventional electrode broadly corresponds with ISRIC criteria (van Reeuwijk,





2002). Similar principles were applied for all soil properties under consideration here; methodological details are provided in Ribeiro et al. (2015).

A next, desired step would be to make the data (e.g. pH, CEC or organic carbon) comparable, 'as if' assessed by a single given (reference) method. That is, fully 'harmonised' and unambiguously defined. However, there is generally no universal

5    equation for converting property values from one method to another in all situations (GlobalSoilMap, 2013; Jankauskas et al., 2006; Lettens et al., 2007). Basically, this implies that within the framework of the Global Soil Partnership (GSP), for example, each regional or continental node will need to develop and apply node-specific conversion functions (towards the yet to be defined GSP-adopted standard reference methods), building on comparative analyses using say archived soil samples (Baritz et al., 2014) and spectral libraries.

## 3 Serving consistent standardised data

WoSIS provides an important building block for the spatial data infrastructure (Figure 3) through which ISRIC WDC-Soils will be serving an increasing range of data (point, raster, polygon) to the international community (Batjes et al., 2013; Hengl et al., 2014). The most recent set of WoSIS-derived data is served via an OGC-compliant WFS (Web Feature Service)

provided by GeoServer instance; the data may be accessed freely via the following webpage: http://www.isric.org/content/wosis-distribution-set. By its nature, however, this dataset will be dynamic as it will grow when additional point data are processed, additional soil attributes are considered, and/or when possible corrections are required.
<< Insert Figure 3 >>

For modelling and citation purposes, we serve *static snapshots* of the standardized data, with clear time stamps, in Comma Separated Values format (CSV). Each *snapshot* will have a unique name and Digital Object Identifier (DOI), for example file *WoSIS_2016_July.zip* with doi: 10.17027/isric-wdcsoils.20160003.

The July 2016 snapshot, as described in this paper, holds data for some 118,400 unique soil profiles of which some 96,000

are georeferenced within defined limits (Figure 4). Overall, the data lineage strongly determined which specific data could be standardized with acceptable confidence (as flagged in the database). The number of measured data for each property varies between profiles and with depth, generally depending on the purpose of the initial studies (see Soil Survey Division Staff, 1993; Landon, 1991). In total, this corresponds with over 31 million records of soil observations and measurements. So far, some 20% thereof have been quality-assessed and 'standardised' according to the present procedures, including both

numeric (20, e.g. sand content) and class (3, e.g. WRB soil classification) properties. Descriptions for the standard variables, naming conventions, and units of measurement are provided in the Appendix.



<< Figure 4 >>

## 4 Towards global soil data interoperability

So far, all datasets managed in WoSIS were provided as 'stand-alone' databases; as such their content is 'static'. Steps are being made towards the development of a federated, and ultimately inter-operable, service or Spatial soil Data Infrastructure (SDI), through which source data are served and updated by the respective data providers and made queryable according to the agreed upon data standards. A first, possible step in this direction -though not yet interoperable- is the exchange of data in a PostgreSQL environment using a Foreign Data Wrapper (FWD). Subsequently, the 'transferred' data can be mapped to the WoSIS data model for further standardisation and harmonisation as described earlier. A technically more challenging solution for the worldwide exchange of soil data was implemented during the OGC Soil Data Interoperability Experiment (soilIE).

SoilIE, undertaken in the second half of 2015, had the objective of developing and testing a soil standard that harmonised existing standards for data exchange defined in Europe and Oceania. During the SoilIE, partners from Europe and Oceania mapped their test data to the SoilML format. Multiple OGC Web Feature Services (WFS) providing data in soilML format were established, allowing for on-line derivation of new data (e.g. using pedotransfer functions). The SoilIE was successful in accessing data in multiple clients (servers) from several soil data providers, each using their own software configurations. Further collaboration will involve refinements to the SoilML schema, Resource Description Foundation (RDF) vocabularies, linked data, and other remaining issues.

Use of OGC web services and modelling data in XML is necessary for fulfilment of compliance requirements with regional interoperability initiatives (INSPIRE, 2015; GS Soil, 2008). The output of the data can then be customized between different XML standards using Extensible Stylesheet Language (XSL) templates or using server schema mapping.

The above activities in support of a global soil SDI were initiated by the GlobalSoilMap consortium in Wageningen, 2009, and may be consolidated within the framework of the Global Soil Partnership (FAO-GSP, 2014a, b; IUSS WG-SIS, 2015) and related interoperability efforts in other domains (e.g. Porter et al., 2015; GEOSS, 2012; GODAN, 2015).

## 5 Data availability

Version WoSIS_2016_July, as described in this paper, is archived for long-term storage at ISRIC – World Soil Information, the World Data Center (WDC) for Soils of the ICSU World Data System; it may be accessed freely through doi: 10.17027/isric-wdcsoils.20160003.



## 6 Conclusions

Bringing disparate soil databases from numerous sources under a common standard poses many and diverse challenges. So far, the focus in WoSIS has been on the standardisation of soil property definitions, soil analytical method descriptions and soil property values in order to serve consistent, quality-assessed data that have been observed or measured according to

5 analytical procedures (aggregates) that are functionally comparable.

Future releases of WoSIS-served data will consider a wider selection of soil site and layer properties, assessed by conventional soil analytical procedures as well as by soil spectroscopy. Further, grid and polygon maps will be gradually added to the server database. Each release (snapshot) will be given a unique time stamp and digital object identifier.

The WoSIS server database forms an important building block of ISRIC's evolving spatial data infrastructure. Instrumental

to enhanced usability of the data served by WoSIS will be the actual harmonization of soil property values as well as the further standardization of identifiers and descriptions of soil analytical procedures. Development of corresponding interfaces will allow for the fulfilment of future exchange of, and demands, for global soil information and enable further processing of soil data shared by contributing parties.

WoSIS-related activities are already catalysing institutional collaboration with institutes in Africa and Latin America.

Capacity building and cooperation among (inter)national soil institutes around the world is essential to create and share ownership of the soil information newly derived from the shared data. Also to strengthen the necessary expertise and capacity to further develop and test the world soil information service worldwide.

### Acknowledgements

The development of WoSIS has been made possible thanks to the contributions and shared knowledge of a steadily growing number of data providers, including soil survey organisations, research institutes and individual experts, whose contributions are gratefully acknowledged. A detailed list of data providers is available at http://www.isric.org/content/wosis-cooperating-institutions-and-experts for details.





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



**Figure captions**

Figure 1. General procedure for processing data in WoSIS

5   Figure 2. Flagging of repeated profiles between the AfSP, ISIS, WISE and SOTER databases (see text for explanation of abbreviations)

Figure 3. Serving consistent soil layers from WoSIS to the user community through ISRIC's evolving spatial data infrastructure

Figure 4. Location op soil profiles considered in WoSIS (July 2016). (See the Appendix for the full complement of
10     properties)



**List of Figures and Tables**

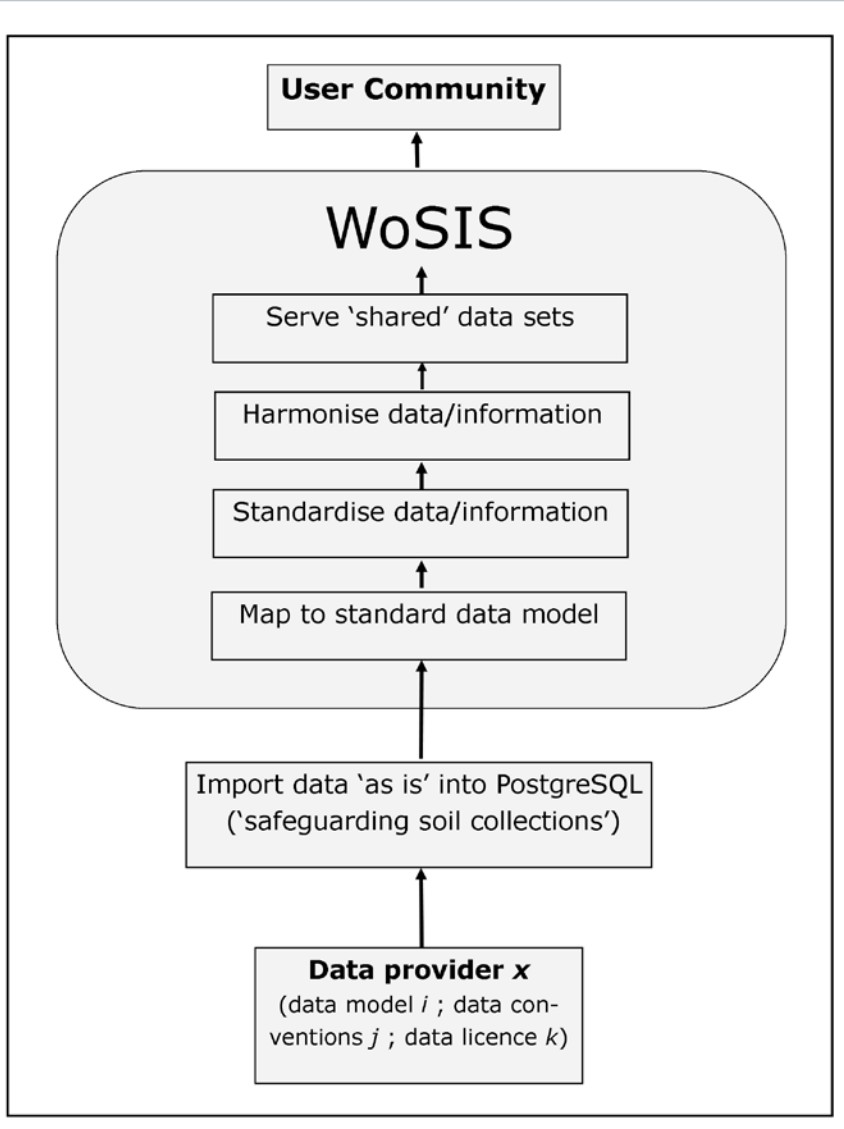

**Figure 1. General procedure for processing data in WoSIS**

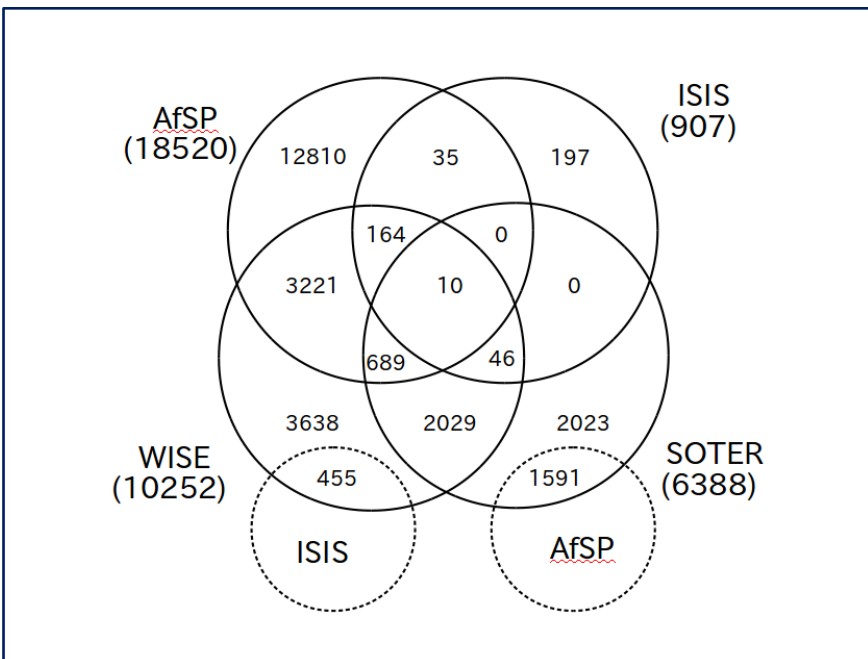

**Figure 2. Flagging of repeated profiles between the AfSP, ISIS, WISE and SOTER databases (see text for explanation of abbreviations)**





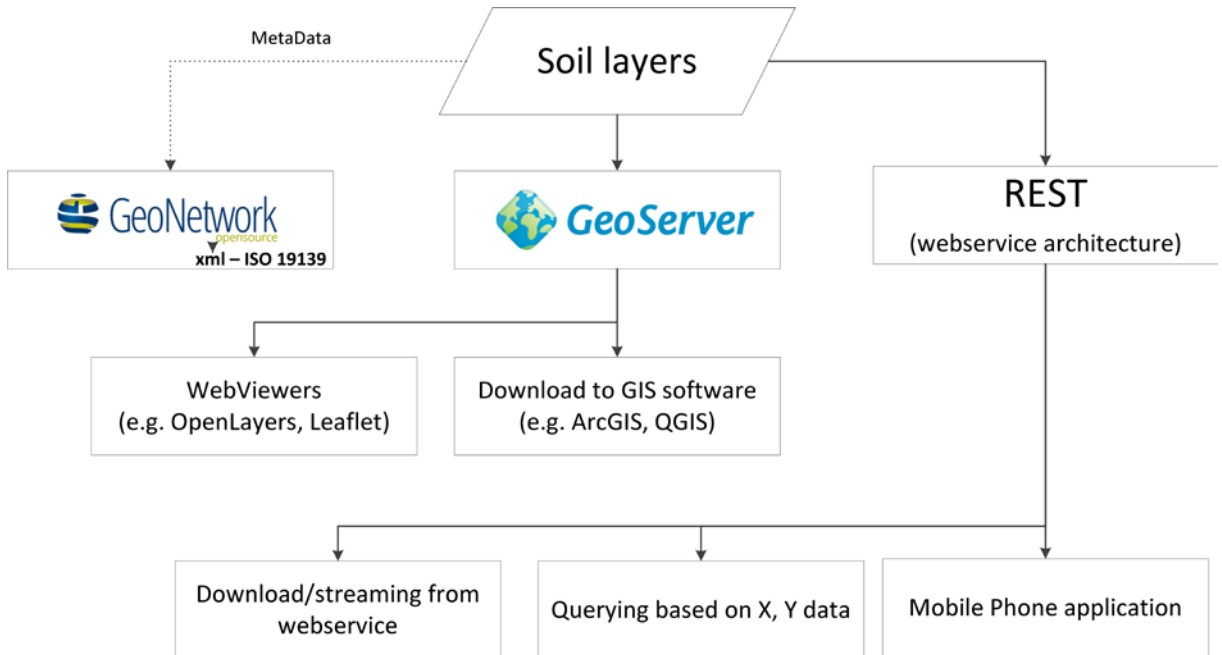

**Figure 3. Serving consistent soil layers from WoSIS to the user community through ISRIC's evolving spatial data infrastructure**







**Figure 4. Location op soil profiles considered in WoSIS (July 2016).**
**(See the Appendix for the full complement of properties)**



## 8 Appendix: Naming conventions and descriptions of variables served from the ISRIC World Soil Information Service

| Name[a] | Description |
| --- | --- |
| bulk_density_fine_earth | Bulk density of the fine earth fraction < 2 mm (kg/dm³) |
| bulk_density_whole_soil | Bulk density of the whole soil including coarse fragments (kg/dm³) |
| calcium_carbonate_equivalent_total | The content of carbonate in a liming material or calcareous soil calculated as if all of the carbonate is in the form of $CaCO_3$ (g/kg in the fine earth fraction < 2 mm); also known as inorganic carbon |
| organic_carbon | Gravimetric content of organic carbon in the fine earth fraction < 2 mm (g/kg) |
| total_carbon | Gravimetric content of organic carbon and inorganic carbon in the fine earth fraction < 2 mm (g/kg) |
| cation_exchange_capacity_pH7 | Capacity of the fine earth fraction < 2 mm to hold exchangeable cations, estimated by buffering the soil at pH7 (CEC, $cmol_c$/kg) |
| cation_exchange_capacity_pH8 | Capacity of the fine earth fraction < 2 mm to hold exchangeable cations, estimated by buffering the soil at pH8.2 (CEC, $cmol_c$/kg) |
| effective_cation_exchange_capacity | Capacity of the fine earth fraction < 2 mm to hold exchangeable cations at the pH of the soil (ECEC, $cmol_c$/kg). Conventionally approximated by summation of exchangeable bases ($Ca^{2+}$, $Mg^{2+}$, $K^+$, and $Na^+$) plus 1 N KCl exchangeable acidity ($Al^{3+}$ and $H^+$) in acidic soils. |
| electrical_conductivity | Ability of a 1:x soil water extract to conduct electrical current ($EC_x$, mS/m); ECe refers to values measured in a saturated soil extract. |
| coarse_fragments_gravimetric_total | Gravimetric content of coarse fragments > 2 mm in the whole soil (g/100g) |
| coarse_fragments_volumetric_total | Volumetric content of coarse fragments > 2 mm in the whole soil ($cm^3$/100$cm^3$) |



| clay_total | Gravimetric content of < 0.002 mm soil material in the fine earth fraction < 2 mm (g/100g) |
|---|---|
| silt_total | 0.002 mm to Y mm fraction of the < 2 mm soil material (g/100g); esd (equivalent spherical diameter), X resp. Y as specified in the analytical method descriptions |
| sand_total | Larger than Y mm fraction of the < 2 mm soil material (g/100g); esd (equivalent spherical diameter) Y as specified in the analytical method descriptions |
| water_retention_gravimetric | Soil moisture content by weight, at the tension specified in the analytical method descriptions (g/100g) |
| water_retention_volumetric | Soil moisture content by volume, at the tension specified in the analytical method descriptions ($cm^3/100cm^3$) |
| ph_cacl2 | A measure of the acidity or alkalinity in soils, defined as the negative logarithm (base 10) of the activity of hydronium ions ($H^+$) in a $CaCl_2$ solution, as specified in the analytical method descriptions |
| ph_h2o | A measure of the acidity or alkalinity in soils, defined as the negative logarithm (base 10) of the activity of hydronium ions ($H^+$) in water |
| ph_kcl | A measure of the acidity or alkalinity in soils, defined as the negative logarithm (base 10) of the activity of hydronium ions ($H^+$) in a KCl solution, as specified in the analytical method descriptions |
| ph_naf | A measure of the acidity or alkalinity in soils, defined as the negative logarithm (base 10) of the activity of hydronium ions (H+) in a NaF solution, as specified in the analytical method descriptions |
| soil_classification_WRB | Classification of the soil profile according to specified edition (year) of the World Reference Base for Soil Resources (WRB), up to qualifier level when available |





| soil_classification_FAO | Classification of the soil profile according to specified edition (year) of the FAO-Unesco Legend, up to soil unit level when available |
|---|---|
| soil_classification_Soil_Taxonomy | Classification of the soil profile according to specified edition (year) of USDA Soil Taxonomy, up to subgroup level when available |

[a] Similar naming conventions are used for the snapshots (see text), but the above names are then supplemented with the data of the snapshot, for example: ph_h2o_2016_July.