# Peer review of "WoSIS: Serving standardised soil profile data for the world"

_Earth System Science Data, 2016_

## Referee Comment (RC1) · G. Hugelius (Referee) · 20 Nov 2016

The WoSIS "Standardized soil profiles for the world" constitutes an immensely valuable dataset that will undoubtedly be used widely and frequently by the scientific and decision-making communities. The dataset as it is presented here builds on many decades of hard work and experience. It is also clear that the data presentation, harmonization and formatting described in this paper follows from a long-process of planning and thinking about how data should be effectively provided to the community (see e.g. Baritz et al., 2014, Leenars et al., 2014 and Ribeiro et al., 2015). As such, I commend the authors for this work and find little reason to suggest major/significant changes or amendments to the present structure of the data. I do have a few suggestion and/or request for clarity (see comments below).

[Figure]

The manuscript is well written, clear and logical in its presentation of the data and the process by which data has been compiled. However, I find that data-users who want more details will likely need to consult the procedures manual of WoSIS (Ribeiro et al 2015). This applies e.g. to how standardization or harmonization of different protocols and methods was carried out (sections 2.6 and 2.7). Given the very wide scope of this work and the length of text required to explain it, I think this is sensible. However, I would suggest that it could be made more clear in the data "read me" file as well as this manuscript where the technical details can be found (as a side-note, the link to the Ribeiro et al (2015) report on http://www.isric.org/data/wosis does not work?).

The only details I would urge the authors to include a fuller presentation and description of in this manuscript is the flags of data quality (i.e. the flags referred to in the methods in section 2 or more specifically on page 6 line 27). How are included and expressed in the database?

I have found that an unexpectedly important variable to make sure a dataset finds the use it deserves from the scientific community is that it is user-friendly and easy to access. Researchers that are highly pressed for time may opt to use older data they have at hand if new data requires too much formatting before use. In light of this my only substantial comment regarding data presentation is that the authors should consider providing one file that contains all data in one large matrix/table. I realize that such a file would likely be too large for e.g. 32-bit MS Excel to handle smoothly, but many users will access the data using other softwares that would have no problem handling it. This would save many users the work of writing individual scripts that combine the different data layers for any individual profile. This will reduce the risk of unintentional processing errors and likely increase the usability of the dataset.

Specific comments

Title: I would consider changing "Serving" to "Providing" which may be a more appropriate term.

Figure1. I find the layout counter-intuitive and would prefer to have it flow the other way around, from top to bottom

I find section 2.4 and figure 2 very assuring with regards to the data quality. I believe that duplication of profile data is a big problem for many databases. With that said, it would be beneficial to data users to get some more details regarding the procedures used to identify these duplicated profiles? Or a clear reference to another source that describes the procedures in detail.

This same concern also applies to section 2.5 which is very brief and would benefit from some more detail. Please define what QA/QC is (Page 4, line 19). I also find the meaning of the subsequent sentence unclear. Does "(first) user" refer to the data provider? In that case, does this reasoning imply that data provided by "trusted sources" was screened differently?

Page 4 line 27. Here the term "pedon" appears. Is this used as a synonym to "soil profile" or does it have a different meaning to the authors?

Sections 2.5 and 2.6 clearly state that "…more details can be found in the procedures manual of the ISRIC World Information Service (Ribeiro et al., 2015)." This is already mentioned in section 2.7.

Page 5 line 19. "In first instance" could be replaced by "In the first generations of WoSIS" or similar, (provided that this is what you mean).

Page 5 line 21. Here, the variables carbon and organic carbon content are not listed. This is different from organic matter content which is listed.

Figure 4 is informative, but it is difficult to judge the quantity of profiles in different regions since there are so many symbols. I believe readers will be interested to get an idea of the total amount of profiles from different regions. I would suggest that you complement figure 4 with a table that summarizes the amount of profiles available from different key geographical regions (e.g. countries with more than a thousand profiles

or continents).

I found section 4 highly interesting and informative, and hopefully coming generations of interoperability may include more geographical regions (notably Asia and the Americas).

---

## Referee Comment (RC2) · A. Samuel-Rosa (Referee) · 29 Nov 2016

ISRIC World Soil Information is long known for its efforts for collecting, organising and serving quality soil data and information for the international community. Its latest enterprise, the World Soil Information Service (WoSIS), comes as the natural outcome of decades of responsible and sound scientific work. This is made very clear in this paper – the reason why I have only a few comments and questions.

The paper successfully describes in a very clear and concise way the procedures adopted by ISRIC for *standardising*, *harmonising* and *serving* global soil profile data. These seem to be the most time consuming in house tasks when building a truly usable soil database and the authors devoted most of their efforts to their description in this paper. However, I think that the paper – and WoSIS – would benefit from more details

on the *safeguarding* step. The way I understand it, this first step of safeguarding soil data involves establishing and maintaining a close and lasting collaboration with soil data providers. Collaboration with institutions from different countries usually involve a series of bureaucratic – and political – decisions. In the case of WoSIS, I believe that data providers would be concerned about issues relating to intellectual property, user rights, and how they can benefit from contributing soil data. Thus, the questions: How does ISRIC approach potential data providers? What are the strategies used by ISRIC to deal with bureaucratic – and political – issues? How can an individual or organization contribute data for inclusion in WoSIS? What immediate benefits one can expect when contributing soil data?

In my experience, the lack of or poor georeferencing of soil profile data is a more serious problem for some soil data users, specially for geospatial applications, than the lack of harmonisation – provided one has the proper means for validating modelling outputs. In the current version of WoSIS, about 20% of the soil profiles do not have a quantitative georeference, i.e. numeric geographic coordinates. This is a hindrance to their usage. Thus, the question: What are the strategies planned for attributing numeric geographic coordinates to soil profiles that lack this sort of information?

Specific comments

Figure 2. Some text is unexpectedly underlined. Consider removing the underline or explaining its usage. Also, consider using a comma as a Thousands separator as it has been used in other parts of the text, e.g. the abstract.

Figure 4 (title): There is a typo in the figure title: "Location op soil profiles [...]" should read "Location of soil profiles [...]".

Figure 4: I suggest reducing the width of points so that the figure gives a more realistic view of under-represented regions of the world. In the current format it gives the false impression that, for example, Latin America (except for Paraguay and parts of Chile and Argentina) is already well represented and Latin Americans should not bother

[Figure]

contributing more data. Also, consider presenting a summary table with the countries with the highest profile density (number of profiles per surface area) in WoSIS.

Finally, I must note that I agree with the comments and notes of Hugelius (2016), and would like to reinforce the suggestion of also serving all data in a singe large CSV file. In this case, for ease of use, I would suggest omitting the metadata, which should only be available in the single soil property files.

Reference

Hugelius, G. Interactive comment on "WoSIS: Serving standardised soil profile data for the world" by Niels H. Batjes et al.. Earth Syst. Sci. Data Discuss. 2016.

---

## Author Comment (AC1) · 13 Dec 2016

**Author response to Referee #1 : ESSDD-2016-34-RC1**

Dear Dr Hugelius,

Thank you for your constructive review and encouraging comments concerning the anticipated usefulness of present and future standardised datasets derived from WoSIS. These comments have helped us to improve the manuscript. In particular, we have followed up on your suggestion to improve the user-friendliness of the WoSIS-derived snapshots, hence the file-format for distributing the data.

Below we address your queries and suggestions in the order that they are raised in the referee report. Revised texts are indicated in blue in the author's comments and accompanying manuscript (which also includes our response to Reviewer 2).

In places, we have reworded some text for enhanced clarity. For example, the phrases concerning the total number of profiles managed in the central WoSIS server database, as of July 2016, versus the number of standardised profile data that we can provide in the 'July 20016' snapshot (see Section 3).

**Referee Comment 1 (RC1):** I find that data-users who want more details will likely need to consult the procedures manual of WoSIS (Ribeiro et al 2015). This applies e.g. to how standardization or harmonization of different protocols and methods was carried out (sections 2.6 and 2.7). Given the very wide scope of this work and the length of text required to explain it, I think this is sensible. However, I would suggest that it could be made more clear in the data "read me" file as well as this manuscript where the technical details can be found (as a side-note, the link to the Ribeiro et al (2015) report on http://www.isric.org/data/wosis does not work?).

**Author comment 1 (AC1):** Thank you for your observation. As recommended, we have expanded the 'ReadMe File.' We have added text to explain the structure of the files served in the snapshot (see details below, under RC3/AC3). We have also added a reference to the technical documentation (procedures manual) for the whole WoSIS database (Ribeiro et al. 2015), which can now be downloaded directly from the new 'ReadMe File' thus enhancing 'user-friendliness'.

The revised "Readme File" is included in the newly formatted version of the "July 2016 snapshot (on-line as of 13/12/2016).

For clarification, section 2.6 and 2.7 now include references to the relevant pages in the procedures manual, for example in Section 2.6: "WoSIS criteria for coding standardised analytical method descriptions are listed in Riberio et al. (2015, p. 47-53).

The typo in the URL has been corrected.

**RC2:** The only details I would urge the authors to include a fuller presentation and description of in this manuscript is the flags of data quality (i.e. the flags referred to in the methods in section 2 or more specifically on page 6 line 27). How are included and expressed in the database?

**AC2**: These flags are processed in the central WoSIS server database, and gradually updated as the various steps of data processing/standardisation proceed (see Figure 2); this type of information is especially important for the database manager (see Ribeiro et al. 2015, p. 90 for table profile, p. 91 for profile_attribute, and p. 93 for profile_layer.attribute). Initially, all data entered/provided are assigned level of trust A for 'As entered, no validation (by ISRIC)'. Upon their standardisation, data are coded as having 'level of trust B'. Ultimately, but only upon full harmonisation, the trust level can be upgraded to level-C (see section 2.7, final paragraph).

For profile locations, we provide a measure for the positional accuracy:
"A measure for the positional accuracy is provided for each profile (e.g., '0.01' when degrees, minutes and seconds are provided, see Ribeiro et al. 2015, p. 90)."

**RC3:** I have found that an unexpectedly important variable to make sure a dataset finds the use it deserves from the scientific community is that it is user-friendly and easy to access. Researchers that are highly pressed for time may opt to use older data they have at hand if new data requires too much formatting before use. In light of this my only substantial comment regarding data presentation is that the authors should consider providing one file that contains all data in one large matrix/table. I realize that such a file would likely be too large for e.g. 32-bit MS Excel to handle smoothly, but many users will access the data using other softwares that would have no problem handling it. This would save many users the work of writing individual scripts that combine the different data layers for any individual profile. This will reduce the risk of unintentional processing errors and likely increase the usability of the dataset.

**AC3:** We greatly value this particular suggestion as user-friendliness is indeed a key factor. The WoSIS team has therefore re-considered the issue: various alternatives to present the standardised data (in addition to the 'dynamic' or latest WFS-set) were explored. The probably best option was then implemented as further described below and in the revised manuscript. Concerning the format we now use tab-delimited TXT-files; this is the most common and widely accepted format that can be opened by any application in any operating system (see ReadMe file in zipped data set).

In brief, we considered four options for grouping and exporting data from WoSIS in TXT format:

1. Flexible way (25 tables)
2. Easy way (23 tables)
3. Everything-in-one file (1 table)
4. New suggestion (3 tables)

Option 1, the 'flexible' way, would be to distribute the data using the minimal number of TXT-files corresponding with tables in a database. This option would give the following tables (files), one TXT-file for unique profiles (through profile_id), another one for layers per profile (through unique profile_id and layer_id) and one TXT-file for each soil property, per soil layer and profile under consideration. Next, these data would have to be joined using the primary keys.

Option 2, the 'easy' way, corresponds to the approach that we have followed in the initial manuscript; there is no need to do any joins in order to see the full information about a certain soil property.

Option 3, 'everything-in-one' file, in our view, brings both disadvantages and some advantages. There will be one large file with many columns. With the anticipated increase in number and type of standard data provided from WoSIS in upcoming snapshots this would probably become 'incomprehensible', thus not user-friendly. Another disadvantage would be that for the less commonly measured soil properties there may be numerous empty records. A possible advantage, is that the columns gid, profile_layer_attribute_id, profile_layer_id, profile_id, dataset_id, license_type, profile_code, country_id, date, top, bottom, geom_accuracy, geom would appear only once and need not be repeated in every TXT file (as is presently the case for the snapshot, see option 2). Then we would just need to add 7 columns (attribute, unit, method, licence, source_database_id, original profile code, and observation date) for each soil property. Such a large file would be "difficult to handle".

Option 4, our 'new 'suggestion', requires 3 tables. The first contains the profile ids 'plus information about the soil classification. The second is a 'long' file listing all soil properties (and descriptors) in sequence. The third file is essentially a compact 'look up' table file with all the codes, descriptions, and units of measurement. The 4-letter coding system is based on SOTER conventions (Van Engelen and Dijkshoorn, 2013), with some modifications necessary to accommodate for the level of detail/standardisation implemented in the WoSIS database.

Upon due deliberation, we consider option 4 to be the recommended solution for serving TXT-snapshots from WoSIS as it minimises repetition and should be readily understood/accessible by most data users. Also the file can readily be imported into SQL databases or accessed from R and similar.

The new format for serving WoSIS-snapshots is described in a new Appendix B, as well as in the revised 'Readme First' file.

**Specific comments:**

**RC4:** Title: I would consider changing "Serving" to "Providing" which may be a more appropriate term.
**AC4**: Agree, this has been done.

**RC5**: Figure1. I find the layout counter-intuitive and would prefer to have it flow the other way around, from top to bottom.
**AC5:** Indeed, this has been done.

**RC6:** It would be beneficial to data users to get some more details regarding the procedures used to identify these duplicated profiles? Or a clear reference to another source that describes the procedures in detail.
**AC6:** The text was modified as follows: "Being derived from multiple data sources, some of which are compilations, there is a risk that the same profiles are imported several times into WoSIS *albeit* using different identifiers. Computerised procedures that consider lineage and geographical proximity checks were developed to screen for possible repeated profiles. The lineage check considers the data source identifiers, uses this information to trace the original data source, and from there looks for duplicates. Alternatively, the proximity check is based on the geographic coordinates. It first identifies profiles that are suspiciously close to another (e.g. < 10 m). Subsequently, the information for these profiles is compared and the database manager assesses the likelihood of such profiles being identical (Ribeiro et al, p. 5-6)."

**RC7:** Please define what QA/QC is (Page 4, line 19). I also find the meaning of the subsequent sentence unclear. Does "(first) user" refer to the data provider? In that case, does this reasoning imply that data provided by "trusted sources" was screened differently?
**AC7**: All data sources are processed using the same, consistent procedures in WoSIS.  The above query has been answered as follows:

"All data sources  are submitted to the same routine QA/QC checks, building on procedures developed for the WISE (Batjes, 1995, p. 52-53) and AfSP (Leenaars, 2012, p. 125-128) databases. For example, this includes checks on referential integrity, data types, geo-location, units of expression, domain ranges, as well as possible 'latitude-longitude inversions' in profile coordinates. It is assumed that the quality requirements of the (first)  data provider are met and that basic quality checks and screening have taken place, this with due consideration for any soil-specific options in the laboratory procedures (Ribeiro et al., 2015)."

**RC8**:  Page 4 line 27. Here the term "pedon" appears. Is this used as a synonym to "soil profile" or does it have a different meaning to the authors?
**AC8:** Profile is meant here, this has been changed.

**RC9**: Sections 2.5 and 2.6 clearly state that:  "… more details can be found in the procedures manual of the ISRIC World Information Service (Ribeiro et al., 2015)." This is already mentioned in section 2.7.
**AC9**: Indeed repetitive; this will be (has been) removed in the upcoming revised manuscript.

**RC10**: Page 5 line 19. "In first instance" could be replaced by "In the first generations of WoSIS" or similar, (provided that this is what you mean).
**AC10**: This has been reworded: "As indicated, in the first version we limited …".

**RC11**: Page 5 line 21. Here, the variables carbon and organic carbon content are not listed.
This is different from organic matter content which is listed.
**AC11**: Changed to organic carbon.

**RC12**: I would suggest that you complement figure 4 with a table that summarizes the amount of profiles available from different key geographical regions (e.g. countries with more than a thousand profiles or continents).

**AC12**: This has been addressed as follows in the text: "The number of profiles per continent is as follows: Africa (17,153), Antarctica (9), Asia (3,089), Europe (1,908), North America (63,077), Oceania (235) and South America (8,970).  Details by continent and country are provided in Appendix C."

**RC13**: I found section 4 highly interesting and informative, and hopefully coming generations of interoperability may include more geographical regions (notably Asia and the Americas).

**AC13:** Thank you for your comment. The next WoSIS-derived snapshot will consider shared data sets for Australia, Cambodia and Canada**.** However, we have not yet been able to process this information as has now been indicated in the text at the end of Section 3. See also the acknowledgment list on our website under **"**2) *Data providers whose contributions still have to be processed in WoSIS"* http://www.isric.org/content/wosis-cooperating-institutions-and-experts.

**WoSIS: Providing standardised soil profile data for the world**

[revised manuscript text omitted]

10 (Fig. 1); technical details are provided elsewhere (Ribeiro et al., 2015).

<< Insert Figure 1 >>

**2.4 Identifying repeated profiles**

15 Being derived from multiple data sources, some of which are compilations, there is a risk that the same profiles are imported several times into WoSIS *albeit* using different identifiers. Computerised procedures that consider lineage and geographical proximity checks were developed to screen for possible repeated profiles. The lineage check considers the data source identifiers, uses this information to trace the original data source, and from there looks for duplicates. Alternatively, the proximity check is based on the geographic coordinates. It first identifies profiles that are suspiciously close to another (e.g.

20 < 10 m). Subsequently, the information for these profiles is compared and the database manager assesses the likelihood of such profiles being identical (Ribeiro et al, p. 5-6). Figure 2 serves to show the results of this time-consuming process for four databases: ISIS (2016), the ISRIC Soil Information System (reference collection); WISE, World Inventory of Soil Emission potentials (compilation, Batjes, 2009); SOTER, Soil and Terrain databases (compilation, Van Engelen, 2011); and AfSP, the Africa Soil Profiles database (compilation, Leenaars, 2013). For example, 12,810 profiles are present only in

25 AfSP, 35 are shared among AfSP and ISIS (the original source), 164 are shared between AfSP, WISE and ISIS, and 10 profiles occur in the four databases. In case of duplicate profiles, all the corresponding data will nonetheless be standardised as described below (i.e. the "flagged" data are maintained in the WoSIS database). However, ultimately, only the profile with the most complete data and detailed lineage will be distributed (see Sect. 3)is maintained for further processing.

30 << Insert Figure  2 >>

**2.5 Basic data quality assessment and control**

all data  are subsequent to the ome  QA/QC checks, building on procedures developed for the WISE (Batjes, 1995, p. 52-53) and AfSP (Leenaars, 2013, p. 125-128) database. For example, this includes checks on referential integrity, data types, geo-location, units of expression, domain ranges, as well as possible "latitude-longitude inversions" in profile coordinates. I is assumed that the quality requirements of the  data provider are met and that basic quality checks and screening have taken place, this with due consideration for any soil-specific options in the laboratory procedures (Ribeiro et al., 2015). This approach allows users of WoSIS-derived data sets to make their own judgement on the quality of individual analytical data, for instance by the assumption that selected data have comparable quality characteristics or an acceptable (inferred) quality compared to their requirements.

**2.6 Standardisation of soil analytical method descriptions**

[revised manuscript text omitted]

**3 Serving consistent standardised data**

The WoSIS server database itself provides an important building block for the spatial data infrastructure (Fig. 3) through 30 which ISRIC WDC-Soils will be serving an increasing range of data (point, raster and polygon) to the international community (Batjes et al., 2013; Hengl et al., 2016). The most recent set of WoSIS-derived point data is served "24/7" via an

OGC-compliant WFS (Web Feature Service) provided by GeoServer instance. These data may be accessed freely via the following webpage: http://www.isric.org/content/wosis-distribution-set. By its nature, however, this dataset will be *dynamic* as it will grow when additional point data are processed, additional soil attributes are considered, and/or when possible corrections are required. Therefore, for consistent modelling and citation purposes, we provide static snapshots of the standardised data with clear time stamps, in tab separated values format. Each snapshot will have a unique name and Digital Object Identifier (DOI), for example file *WoSIS_2016_July.zip* with doi: 10.17027/isric-wdcsoils.20160003.

<< Insert Figure 3 >>

At the time of writing, the WoSIS server database contained some 118,400 unique "shared" soil profiles, out of which over 96,000 are georeferenced within defined limits, corresponding with some 31 million soil records in total. So far, some 20% thereof have been quality-assessed and standardised using the sequential procedure discussed in this paper. As indicated, the number of measured data for each property varies between profiles and with depth, generally depending on the purpose of the initial studies. Therefore, the data lineage strongly determined which specific data could be served with acceptable confidence (as flagged in the central WoSIS database, see Ribeiro et al. 2015, p. 92). By implication, the "July 2016" snapshot only includes standardised data for 94,441 geo-referenced profiles, representing some 455 thousand layers (or horizons). In total, this corresponds with over 4 million records that include both numeric (19, e.g. sand content or soil pH) as well as class (3, e.g. WRB soil classification) properties. The naming conventions, units of measurement, and file structure are described in Appendix A and B, respectively.

The number of profiles per continent is highest for North America (63,077) and Africa (17,153), followed by South America (8,970), Asia (3,089), Europe (1,908), Oceania (235), and Antarctica (9). These profiles come from 148 countries; the average density of observations is 0.7 profiles per 1000 km$^2$. The actual density of observations varies greatly, both between countries (Appendix C) and within each country (Fig. 4). Such geographic gaps will be filled gradually in the future, this largely depending though on the willingness or ability of data providers to freely share (some of) their data for consideration in WoSIS. Alternatively, it should be noted here that some recently shared datasets are not yet included in the present snapshot (e.g. Australia, Canada and Cambodia).

<< Insert Figure 4 >>

**4 Towards global soil data interoperability**

[revised manuscript text omitted]

**Apppendix A: Naming conventions and descriptions of variables provided in the "WOSIS July 2016" snapshot**

| Code [a] | Attribute | Unit | Profiles | Layers | Description |
|---|---|---|---|---|---|
| BDFI | Bulk density fine earth | kg dm$^{-3}$ | 20 727 | 105 848 | Bulk density of the fine earth fraction < 2 mm (kg/dm$^3$) |
| BDWS | Bulk density whole soil | kg dm$^{-3}$ | 25 909 | 153 568 | Bulk density of the whole soil including coarse fragments (kg/dm$^3$) |
| TCEQ | Calcium carbonate equivalent total | g kg$^{-1}$ | 27 809 | 115 448 | The content of carbonate in a liming material or calcareous soil calculated as if all of the carbonate is in the form of $CaCO_3$ (g/kg in the fine earth fraction < 2 mm); also known as inorganic carbon |
| CECX | Cation exchange capacity (CEC) | cmol$_{(c)}$ kg$^{-1}$ | 48 461 | 273 346 | Capacity of the fine earth fraction < 2 mm to hold exchangeable cations, estimated by buffering the soil at specified pH (e.g. pH7 or pH8; cmol$_c$/kg) |
| CLAY | Clay total | g kg$^{-1}$ | 80 082 | 408 452 | Gravimetric content of < 0.002 mm soil material in the fine earth fraction < 2 mm (g/100g) |
| CFGR | Coarse fragments gravimetric total | 10$^{-2}$ g g$^{-1}$ | 27 050 | 159 206 | Gravimetric content of coarse fragments > 2 mm in the whole soil (g/100g) |
| CFVO | Coarse fragments volumetric total | 10$^{-2}$ cm³ cm$^{-3}$ | 37 280 | 198 534 | Volumetric content of the coarse fragments > 2 mm in the whole soil (cm$^3$/100cm$^3$) |
| ECEC | Effective cation exchange capacity (ECEC) | cmol$_{(c)}$ kg$^{-1}$ | 23 189 | 102 665 | Capacity of the fine earth fraction < 2 mm to hold exchangeable cations at the pH of the soil (ECEC, cmol$_c$/kg). Conventionally approximated by summation of exchangeable bases ($Ca^{2+}$, $Mg^{2+}$, $K^+$, and $Na^+$) plus 1 N KCl exchangeable acidity ($Al^{3+}$ and $H^+$) in acidic soils |
| ELCO | Electrical conductivity | dS m$^{-1}$ | 28 266 | 120 039 | Ability of a 1:x soil water extract to conduct electrical current ($EC_x$, dS/m); $EC_e$ refers to values measured in a saturated soil extract |
| ORGC | Organic carbon | g kg$^{-1}$ | 64 118 | 315 362 | Gravimetric content of organic carbon in the fine earth fraction |
| PHCA | pH CaCl$_2$ | unitless | 39 074 | 237 756 | A measure of the acidity or alkalinity in soils, defined as the negative logarithm (base 10) of the activity of hydronium ions ($H^+$) in a $CaCl_2$ solution, as specified in the analytical method descriptions (dimensionless) |
| PHAQ | pH H$_2$O | unitless | 79 118 | 407 226 | A measure of the acidity or alkalinity in soils, defined as the negative logarithm (base 10) of the activity of hydronium ions ($H^+$) in water (dimensionless) |
| PHKC | pH KCl | unitless | 19 064 | 88 127 | A measure of the acidity or alkalinity in soils, defined as the negative logarithm (base 10) of the activity of hydronium ions ($H^+$) in a KCl solution, as specified in the analytical method descriptions (dimensionless) |
| PHNF | pH NaF | unitless | 4866 | 24 917 | A measure of the acidity or alkalinity in soils, defined as the negative logarithm (base 10) of the activity of hydronium ions ($H^+$) in a NaF solution, as specified in the analytical method descriptions (dimensionless) |
| SAND | Sand total | 10$^{-2}$ g g$^{-1}$ | 78 402 | 398 573 | Larger than Y mm fraction of the < 2 mm soil material (g/100g); esd (equivalent spherical diameter), Y as specified in the analytical method descriptions |
| SILT | Silt total | 10$^{-2}$ g g$^{-1}$ | 79 331 | 406 502 | 0.002 mm to Y mm fraction of the < 2 mm soil material (g/100g); esd (equivalent spherical diameter), X respectively Y as specified in the analytical method descriptions |

| Code [a] | Attribute | Unit | Profiles | Layers | Description |
|---|---|---|---|---|---|
| CFAO | Soil classification FAO | unitless | 24 894 | 24 894 | Classification of the soil profile according to specified edition (year) of the FAO-Unesco Legend, up to soil unit level when available |
| CSTX | Soil classification Soil taxonomy | unitless | 21 614 | 21 614 | Classification of the soil profile according to specified edition (year) of USDA Soil Taxonomy, up to subgroup level when available |
| CWRB | Soil classification WRB | unitless | 24 628 | 24 628 | Classification of the soil profile according to specified edition (year) of the World Reference Base for Soil Resources (WRB), up to qualifier level when available |
| TOTC | Total carbon | $g\ kg^{-1}$ | 14 094 | 70 687 | Gravimetric content of organic carbon and inorganic carbon in the fine earth fraction < 2 mm (g/kg) |
| WRGR | Water retention gravimetric | $10^{-2}\ g\ g^{-1}$ | 28 701 | 173 972 | Soil moisture content by weight, at the tension specified in the analytical method descriptions (g/100g) |
| WRVO | Water retention volumetric | $10^{-2}\ cm^3\ cm^{-3}$ | 17 124 | 82 695 | Soil moisture content by volume, at the tension specified in the analytical method descriptions ($cm^3/100cm^3$) |

[a]    A full complement of all these attributes is generally not available for many profiles (see text).

**Appendix B: Structure of the "July 2016" WoSIS snapshot**

This Appendix describes the structure of the data files presented in the "July 2016" WoSIS snapshot:

- *wosis_201607_attributes.txt*
- *wosis_201607_profiles.txt*, and
- *wosis_201607_layers.txt*.

The first file lists the four letter code for each attribute, a short explanation, and the units of measurement, respectively (Appendix A). This file also gives the number of profiles and layers in the present snapshot.

The second file lists the unique profile ID (i.e. primary key), country name and ISO country code, geometric accuracy, latitude and longitude (WGS 1984) as well as information on the soil classification system and edition. Depending on the soil classification system used, the number of fields will vary. For example, for US Soil Taxonomy, coded here as "cstx", these are: order, suborder, great group and subgroup as indicated in the column headings.

The third, largest file, lists all the soil properties by layer and profile to enhance user-friendliness. It starts with:

| | |
|---|---|
| profile_id | identifier for profile, links to file wosis_201607_profiles. |
| profile_layer_id | unique identifier for layer for given profile (primary key) |
| top | upper depth of layer (or horizon) |
| bottom | lower depth of layer |

Subsequently, the following items are listed sequentially per attribute ("xxxx") as defined under "code" in file *wosis_201607_attributes.txt*:

| | |
|---|---|
| xxxx_value | array listing all values for soil property "xxxx" for the given layer; thus, more than one observation can be reported when available, for example 3 values for ORGC: {1:0.55, 2:1.01, 3:0.85} |
| xxxx_value_avg | average, for above (use this value for "routine" modelling) |
| xxxx_method | array listing the method descriptions for each value |
| xxxx_date | array listing the date of observation for each value |
| xxxx_dataset_id | abbreviation for source data set (e.g. WD-ISIS), |
| xxxx_profile_code | code for given profile |
| xxxx_licenset | licence for given data (CC-BY-NC or CC-BY) |

(... )                          as above, but for the next attribute

All fields in the above files are tab-delimited, while double quotation marks serve as text delimiters; file coding is according to the UTF-8 unicode transformation format. As such, the files can be easily imported into an SQL database or statistical software such as R, after which they may be joined using the unique profile_id.

**Appendix C: Number of profiles by country and continent.**

| Continent | Country_name | ISO code | N of profiles | Area (km$^2$) | Profile density (per 1000 km$^2$) |
|---|---|---|---|---|---|
| Africa | Algeria | DZ | 4 | 2 308 647 | 0.002 |
| | Angola | AO | 1035 | 1 246 690 | 0.830 |
| | Benin | BJ | 738 | 115 247 | 6.404 |
| | Botswana | BW | 898 | 578 247 | 1.553 |
| | Burkina Faso | BF | 887 | 273 281 | 3.246 |
| | Burundi | BI | 36 | 26 857 | 1.340 |
| | Cameroon | CM | 455 | 465 363 | 0.978 |
| | Central African Republic | CF | 87 | 619 591 | 0.140 |
| | Chad | TD | 5 | 1 265 392 | 0.004 |
| | Congo | CG | 70 | 340 599 | 0.206 |
| | Côte d'Ivoire | CI | 254 | 321 762 | 0.789 |
| | Dem. Rep. of the Congo | CD | 374 | 2 329 162 | 0.161 |
| | Egypt | EG | 22 | 98 2161 | 0.022 |
| | Ethiopia | ET | 1583 | 1 129 314 | 1.402 |
| | Gabon | GA | 46 | 264 022 | 0.174 |
| | Ghana | GH | 163 | 238 842 | 0.682 |
| | Guinea | GN | 62 | 243 023 | 0.255 |
| | Guinea-Bissau | GW | 18 | 30 740 | 0.586 |
| | Kenya | KE | 504 | 582 342 | 0.865 |
| | Lesotho | LS | 33 | 30 453 | 1.084 |
| | Liberia | LR | 48 | 96 103 | 0.499 |
| | Libya | LY | 14 | 1 620 583 | 0.009 |
| | Madagascar | MG | 52 | 588 834 | 0.088 |
| | Malawi | MW | 2985 | 118 715 | 25.144 |
| | Mali | ML | 756 | 1 251 471 | 0.604 |
| | Mauritania | MR | 11 | 1 038 527 | 0.011 |
| | Morocco | MA | 27 | 414 030 | 0.065 |
| | Mozambique | MZ | 275 | 787 305 | 0.349 |
| | Namibia | NA | 62 | 823 989 | 0.075 |
| | Niger | NE | 488 | 1 182 602 | 0.413 |
| | Nigeria | NG | 1203 | 908 978 | 1.323 |
| | Rwanda | RW | 92 | 25 388 | 3.624 |
| | Senegal | SN | 311 | 196 200 | 1.585 |
| | Sierra Leone | SL | 12 | 72 281 | 0.166 |
| | Somalia | SO | 245 | 632 562 | 0.387 |
| | South Africa | ZA | 649 | 1 220 127 | 0.532 |
| | Sudan | SD | 116 | 1 843 196 | 0.063 |
| | Swaziland | SZ | 14 | 17 290 | 0.810 |
| | Togo | TG | 9 | 56 767 | 0.159 |
| | Tunisia | TN | 60 | 155 148 | 0.387 |
| | Uganda | UG | 12 | 241 495 | 0.050 |

| Continent | Country_name | ISO code | N of profiles | Area (km$^2$) | Profile density (per 1000 km$^2$) |
|-----------|--------------|----------|---------------|---------------|-----------------------------------|
| | United Republic of Tanzania | TZ | 1647 | 939 588 | 1.753 |
| | Zambia | ZM | 472 | 751 063 | 0.628 |
| | Zimbabwe | ZW | 319 | 390 648 | 0.817 |
| Antarctica | Antarctica | AQ | 9 | 12 537 967 | 0.001 |
| Asia | Afghanistan | AF | 19 | 641 827 | 0.030 |
| | Armenia | AM | 6 | 29 624 | 0.203 |
| | Azerbaijan | AZ | 4 | 164 780 | 0.024 |
| | Bahrain | BH | 2 | 673 | 2.970 |
| | Bangladesh | BD | 16 | 139 825 | 0.114 |
| | Bhutan | BT | 80 | 37 674 | 2.123 |
| | China | CN | 1490 | 9 345 214 | 0.159 |
| | Georgia | GE | 9 | 69 785 | 0.129 |
| | India | IN | 139 | 2 961 118 | 0.047 |
| | Indonesia | ID | 108 | 1 888 620 | 0.057 |
| | Iran (Islamic Republic of) | IR | 2 | 1 677 319 | 0.001 |
| | Iraq | IQ | 14 | 435 864 | 0.032 |
| | Israel | IL | 16 | 20 720 | 0.772 |
| | Japan | JP | 39 | 373 651 | 0.104 |
| | Jordan | JO | 40 | 89 063 | 0.449 |
| | Lebanon | LB | 6 | 10 136 | 0.592 |
| | Malaysia | MY | 46 | 329 775 | 0.139 |
| | Mongolia | MN | 7 | 1 564 529 | 0.004 |
| | Nepal | NP | 141 | 147 437 | 0.956 |
| | Oman | OM | 9 | 308 335 | 0.029 |
| | Pakistan | PK | 43 | 788 439 | 0.055 |
| | Philippines | PH | 68 | 296 031 | 0.230 |
| | Republic of Korea | KR | 17 | 99 124 | 0.172 |
| | Sri Lanka | LK | 13 | 66 173 | 0.196 |
| | Syrian Arab Republic | SY | 66 | 188 128 | 0.351 |
| | Taiwan | TW | 33 | 36 127 | 0.913 |
| | Tajikistan | TJ | 5 | 142 004 | 0.035 |
| | Thailand | TH | 285 | 515 417 | 0.553 |
| | Turkey | TR | 68 | 781 229 | 0.087 |
| | United Arab Emirates | AE | 6 | 71 079 | 0.084 |
| | Uzbekistan | UZ | 8 | 449 620 | 0.018 |
| | Yemen | YE | 284 | 453 596 | 0.626 |
| Europe | Albania | AL | 63 | 28 682 | 2.197 |
| | Belarus | BY | 94 | 207 581 | 0.453 |
| | Belgium | BE | 190 | 30 669 | 6.195 |
| | Bulgaria | BG | 45 | 111 300 | 0.404 |
| | Czech Republic | CZ | 38 | 78 845 | 0.482 |

| Continent | Country_name | ISO code | N of profiles | Area (km$^2$) | Profile density (per 1000 km$^2$) |
|---|---|---|---|---|---|
| | Denmark | DK | 20 | 44 458 | 0.450 |
| | Estonia | EE | 123 | 45 441 | 2.707 |
| | Finland | FI | 24 | 336 892 | 0.071 |
| | France | FR | 53 | 548 785 | 0.097 |
| | Germany | DE | 51 | 357 227 | 0.143 |
| | Greece | GR | 11 | 132 549 | 0.083 |
| | Hungary | HU | 61 | 93 119 | 0.655 |
| | Iceland | IS | 11 | 102 566 | 0.107 |
| | Ireland | IE | 36 | 69 809 | 0.516 |
| | Italy | IT | 86 | 301 651 | 0.285 |
| | Latvia | LV | 10 | 64 563 | 0.155 |
| | Lithuania | LT | 18 | 64 943 | 0.277 |
| | Luxembourg | LU | 128 | 2621 | 48.842 |
| | Netherlands | NL | 192 | 35 203 | 5.454 |
| | Norway | NO | 10 | 324 257 | 0.031 |
| | Poland | PL | 128 | 311 961 | 0.410 |
| | Portugal | PT | 35 | 91 876 | 0.381 |
| | Republic of Moldova | MD | 32 | 33 798 | 0.947 |
| | Romania | RO | 43 | 238 118 | 0.181 |
| | Russian Federation | RU | 156 | 16 998 830 | 0.009 |
| | Slovakia | SK | 40 | 49 072 | 0.815 |
| | Spain | ES | 42 | 505 752 | 0.083 |
| | Sweden | SE | 26 | 449 212 | 0.058 |
| | Switzerland | CH | 10 | 41 257 | 0.242 |
| | Ukraine | UA | 79 | 600 526 | 0.132 |
| | United Kingdom | GB | 53 | 244 308 | 0.217 |
| North America | Barbados | BB | 3 | 433 | 6.928 |
| | Belize | BZ | 21 | 21 764 | 0.965 |
| | Canada | CA | 148 | 9 875 646 | 0.015 |
| | Costa Rica | CR | 55 | 51 042 | 1.078 |
| | Cuba | CU | 52 | 110 863 | 0.469 |
| | Dominican Republic | DO | 1 | 48 099 | 0.021 |
| | El Salvador | SV | 9 | 20 732 | 0.434 |
| | Greenland | GL | 5 | 2 165159 | 0.002 |
| | Guatemala | GT | 22 | 109 062 | 0.202 |
| | Honduras | HN | 11 | 112 124 | 0.098 |
| | Jamaica | JM | 76 | 10 965 | 6.931 |
| | Mexico | MX | 12 223 | 1 949 527 | 6.270 |
| | Netherlands Antilles | AN | 4 | 790 | 5.066 |
| | Nicaragua | NI | 26 | 128 376 | 0.203 |
| | Panama | PA | 25 | 74 850 | 0.334 |
| | Puerto Rico | PR | 30 | 8937 | 3.357 |
| | Trinidad and Tobago | TT | 2 | 5144 | 0.389 |

| Continent | Country_name | ISO code | N of profiles | Area (km$^2$) | Profile density (per 1000 km$^2$) |
|---|---|---|---|---|---|
| | United States of America | US | 50 361 | 9 315 946 | 5.406 |
| | United States Virgin Islands | VI | 3 | 352 | 8.514 |
| Oceania | Australia | AU | 142 | 7 687 634 | 0.018 |
| | Cook Islands | CK | 1 | 241 | 4.142 |
| | Fiji | FJ | 9 | 18 293 | 0.492 |
| | Micronesia (Feder. States of) | FM | 14 | 740 | 18.917 |
| | New Zealand | NZ | 20 | 270 415 | 0.074 |
| | Papua New Guinea | PG | 31 | 462 230 | 0.067 |
| | Samoa | WS | 17 | 2835 | 5.996 |
| | Solomon Islands | SB | 1 | 28 264 | 0.035 |
| South America | Argentina | AR | 238 | 2 780 175 | 0.086 |
| | Bolivia (Plurinational State of) | BO | 77 | 1 084 491 | 0.071 |
| | Brazil | BR | 7842 | 8 485 946 | 0.924 |
| | Chile | CL | 45 | 753 355 | 0.060 |
| | Colombia | CO | 166 | 1 137 939 | 0.146 |
| | Ecuador | EC | 77 | 256 249 | 0.300 |
| | French Guiana | GF | 7 | 83 295 | 0.084 |
| | Guyana | GY | 43 | 211 722 | 0.203 |
| | Peru | PE | 147 | 1 290 640 | 0.114 |
| | Suriname | SR | 27 | 145 100 | 0.186 |
| | Uruguay | UY | 131 | 177 811 | 0.737 |
| | Venezuela (Bolivarian Rep. of) | VE | 170 | 912 025 | 0.186 |
| World | World (total) | WD | 94 441 | 137 770 610 | 0.685 |

Note: Country names and areas are based on the Global Administrative Layers (GAUL) database, see:

http://www.fao.org/geonetwork/srv/en/metadata.show?id=12691.

**Acknowledgements**

The development of WoSIS has been made possible thanks to the contributions and shared knowledge of a steadily growing number of data providers, including soil survey organisations, research institutes and individual experts, whose contributions are gratefully acknowledged. A detailed list of data providers is available at http://www.isric.org/content/wosis-cooperating-institutions-and-experts for details.

40

**Figure captions**

**Figure 1**. General procedure for processing data in WoSIS

5   **Figure 2.** Flagging of repeated profiles between the AfSP, ISIS, WISE and SOTER databases (see text for explanation of abbreviations)

F**igure 3.** Serving consistent soil layers from WoSIS to the user community through ISRIC's evolving spatial data infrastructure

**Figure 4.** Location of soil profiles provided in the "July 2016" snapshot of WoSIS.
10   (See Appendix C for the number of profiles by country)

[Figure]

**Figure 1.** General procedure for processing data in WoSIS

[Figure]

**Figure 2**. Flagging of repeated profiles between the AfSP, ISIS, WISE and SOTER databases (see text for explanation of abbreviations)

[Figure]

**Figure 3.** Serving consistent soil layers from WoSIS to the user community through ISRIC's evolving spatial data
infrastructure

[Figure]

**Figure 4**. Location of soil profiles provided in the "July 2016" snapshot of  WoSIS.
(See Appendix C for the number of profiles by country)

---

## Author Comment (AC2) · 13 Dec 2016

**Author response to Referee #2 : ESSDD-2016-34-RC1**

Dear Dr Samuel-Rosa,

Thank you for your kind words about ISRIC's long-term role as trusted broker of quality-assessed soil information and your constructive review.

Below we address your queries and suggestions in the order that they are raised in the referee report. Revised texts are indicated in blue in the author's comments and accompanying adapted manuscript (which also includes our response to Reviewer 1).

**Referee Comment 1 (RC1):** How does ISRIC approach potential data providers? What are the strategies used by ISRIC to deal with bureaucratic – and political – issues? How can an individual or organization contribute data for inclusion in WoSIS? What immediate benefits one can expect when contributing soil data?

**Author Response (AR1):** These are indeed important observations. We now realise that we had omitted to add a reference to ISRIC's Data Policy, which outlines the terms under which data providers may share their data with ISRIC and how these contributions/materials should be acknowledged and cited by data users. We also added some text outlining how potential data providers may contribute data to the WoSIS effort (see Section 2.1): "Everyone may contribute data for inclusion in WoSIS. Data may be submitted in various ways. Analogue data should be provided using a template with standardised variable names as described in the WoSIS Procedures Manual (Ribeiro et al., 2015, p. 378-40). Alternatively, large digital data sets comprising over thousand profiles can be provided to ISRIC as zip files containing the database, documentation and metadata. Prior to any data processing at ISRIC, the data provider must agree in writing with the terms of the Data Policy (ISRIC, 2016)."

Some (fairly) immediate benefits data providers may expect will be an update of the SoilGrids soil property and class maps for their country or region, once a new set of SoilGrids is released. Also, we provide training and capacity building on methodologies, provided there are external project funds for this. Many potential data providers and data users attend our annual Spring School, which may provide the basis for a guest researchership at our institute.

So far, potential data contributors have mainly been contacted directly using our international network. We are addressing the above mentioned 'bureaucracy and political issues' mainly through participation in larger networks, that operate at the "supra-national" level, such as the Global Soil Partnership and GODAN that are working towards free and open data. A discussion of these issues, however, we consider beyond the scope of this data paper (see e.g. http://www.isric.org/sites/default/files/ISRIC_Report_2013_05_LR.pdf).

**RC2**: In the current version of WoSIS, about 20% of the soil profiles do not have a quantitative georeference, i.e. numeric geographic coordinates. This is a hindrance to their usage. Thus, the question: What are the strategies planned for attributing numeric geographic coordinates to soil profiles that lack this sort of information?

**AC2:** Of course, we would prefer have only georeferenced data in the WoSIS database. However, some legacy data come without adequate geographic information to accurately pinpoint their location. Realistically, filling such gaps remains the responsibility of the data providers as they best know their resources; this could become an important task in the framework of the Global Soil Partnership, which could develop an on-line platform for doing this. Once the coordinates are provided together with the original profile identifier, they can be easily accommodated in the WoSIS database. That being said, profiles that lack coordinates, though not suited for digital soil mapping purposes, can be very useful for traditional soil mapping purposes and to derive pedotransfer and taxotransfer functions.

**Specific comments:**

**RC3**: Figure 2. Some text is unexpectedly underlined. Consider removing the underline or explaining its usage. Also, consider using a comma as a Thousands separator as it has been used in other parts of the text, e.g. the abstract.
**AC3**: Thanks, done.

**RC4**: Figure 4 (title): There is a typo in the figure title: "Location op soil profiles [...]" should read "Location of soil profiles [...]".

**AC4**: Thanks, done.

**RC5**: Figure 4: I suggest reducing the width of points so that the figure gives a more realistic view of under-represented regions of the world. In the current format it gives the false impression that, for example, Latin America (except for Paraguay and parts of Chile and Argentina) is already well represented and Latin Americans should not bother contributing more data. Also, consider presenting a summary table with the countries with the highest profile density (number of profiles per surface area) in WoSIS.
**AC5**: We have addressed this point by adding profile counts per continent (Section 3) and by country (see Appendix C, with number of profiles and profile densities). The text has been adapted:

"The number of profiles per continent is highest for North America (63,077) and Africa (17,153), followed by South America (8,970), Asia (3,089), Europe (1,908), Oceania (235), and Antarctica (9). These profiles come from 148 countries; the average density of observations is 0.7 profiles per 1000 km$^2$. The actual density of observations varies greatly, both between countries (Appendix C) and within each country (Fig. 4). Such geographic gaps will be filled gradually in the future, this largely depending though on the willingness or ability of data providers to freely share (some of) their data for consideration in WoSIS. Alternatively, it should be

noted here that some recently shared datasets are not yet included in the present snapshot (e.g. Australia, Canada and Cambodia)."

**RC6:** Finally, I must note that I agree with the comments and notes of Hugelius (2016), and would like to reinforce the suggestion of also serving all data in a single large CSV file. In this case, for ease of use, I would suggest omitting the metadata, which should only be available in the single soil property files.

**AC6:** As indicated in our reply to Gustaf Hugelius (Referee 1), we have considered this issue very carefully. The "snapshot" data are now presented in a format that should be much more user friendly, namely using three txt files that can be easily uploaded into an SQL database or similar (R software) for further use. For details on the file structure please see our response to the Dr Hugelius, Appendix B, as well as the revised 'Readme First' file that comes with the zipped snapshot.

**WoSIS: ProvidingServing standardised soil profile data for the world**

[revised manuscript text omitted]

<< Insert Figure 1 >>

**2.4 Identifying repeated profiles**

Being derived from multiple data sources, some of which are compilations, there is a risk that the same profiles are imported several times into WoSIS *albeit* using different identifiers. Computerised procedures that consider lineage and geographical proximity checks were developed to screen for possible repeated profiles. The lineage check considers the data source identifiers, uses this information to trace the original data source, and from there looks for duplicates. Alternatively, the proximity check is based on the geographic coordinates. It first identifies profiles that are suspiciously close to another (e.g. < 10 m). Subsequently, the information for these profiles is compared and the database manager assesses the likelihood of such profiles being identical (Ribeiro et al, p. 5-6). Figure 2 serves to show the results of this time-consuming process for four databases: ISIS (2016), the ISRIC Soil Information System (reference collection); WISE, World Inventory of Soil Emission potentials (compilation, Batjes, 2009); SOTER, Soil and Terrain databases (compilation, Van Engelen, 2011); and AfSP, the Africa Soil Profiles database (compilation, Leenaars, 2013). For example, 12,810 profiles are present only in AfSP, 35 are shared among AfSP and ISIS (the original source), 164 are shared between AfSP, WISE and ISIS, and 10 profiles occur in the four databases. In case of duplicate profiles, all the corresponding data will nonetheless be standardised as described below (i.e. the "flagged" data are maintained in the WoSIS database). However, ultimately, only the profile with the most complete data and detailed lineage will be distributed (see Sect. 3)is maintained for further processing.

<< Insert Figure  2 >>

**2.5 Basic data quality assessment and control**

For Aall data sources data  are submittedsubsequent to the saome basic QA/QC checks, building on procedures developed for the WISE (Batjes, 1995, p. 52-53) and AfSP (Leenaars, 2013, p. 125-128) database. For example, this includes checks on referential integrity, data types, geo-location, units of expression, domain ranges, as well as possible "latitude-longitude inversions" in profile coordinates. I it is assumed that the quality requirements of the (first) user data provider are met and that basic quality checks and screening have taken place, this with due consideration for any soil-specific options in the laboratory procedures (Ribeiro et al., 2015). This approach allows users of WoSIS-derived data sets to make their own judgement on the quality of individual analytical data, for instance by the assumption that selected data have comparable quality characteristics or an acceptable (inferred) quality compared to their requirements.

**2.6 Standardisation of soil analytical method descriptions**

[revised manuscript text omitted]

25   yet to be defined GSP-adopted standard reference methods, see Baritz et al. 2014), building on comparative analyses using say archived soil samples and spectral libraries.

**3 Serving consistent standardised data**

The WoSIS server database itself provides an important building block for the spatial data infrastructure (Fig. 3) through

30   which ISRIC WDC-Soils will be serving an increasing range of data (point, raster and polygon) to the international community (Batjes et al., 2013; Hengl et al., 2016). The most recent set of WoSIS-derived point data is served "24/7" via an

OGC-compliant WFS (Web Feature Service) provided by GeoServer instance. These data may be accessed freely via the following webpage: http://www.isric.org/content/wosis-distribution-set. By its nature, however, this dataset will be *dynamic* as it will grow when additional point data are processed, additional soil attributes are considered, and/or when possible corrections are required. Therefore, for consistent modelling and citation purposes, we provide static snapshots of the standardised data with clear time stamps, in tab separated values format. Each snapshot will have a unique name and Digital Object Identifier (DOI), for example file *WoSIS_2016_July.zip* with doi: 10.17027/isric-wdcsoils.20160003.

<< Insert Figure 3 >>

At the time of writing, the WoSIS server database contained some 118,400 unique "shared" soil profiles, out of which over 96,000 are georeferenced within defined limits, corresponding with some 31 million soil records in total. So far, some 20% thereof have been quality-assessed and standardised using the sequential procedure discussed in this paper. As indicated, the number of measured data for each property varies between profiles and with depth, generally depending on the purpose of the initial studies. Therefore, the data lineage strongly determined which specific data could be served with acceptable confidence (as flagged in the central WoSIS database, see Ribeiro et al. 2015, p. 92). By implication, the "July 2016" snapshot only includes standardised data for 94,441 geo-referenced profiles, representing some 455 thousand layers (or horizons). In total, this corresponds with over 4 million records that include both numeric (19, e.g. sand content or soil pH) as well as class (3, e.g. WRB soil classification) properties. The naming conventions, units of measurement, and file structure are described in Appendix A and B, respectively.

The number of profiles per continent is highest for North America (63,077) and Africa (17,153), followed by South America (8,970), Asia (3,089), Europe (1,908), Oceania (235), and Antarctica (9). These profiles come from 148 countries; the average density of observations is 0.7 profiles per 1000 km$^2$. The actual density of observations varies greatly, both between countries (Appendix C) and within each country (Fig. 4). Such geographic gaps will be filled gradually in the future, this largely depending though on the willingness or ability of data providers to freely share (some of) their data for consideration in WoSIS. Alternatively, it should be noted here that some recently shared datasets are not yet included in the present snapshot (e.g. Australia, Canada and Cambodia).

<< Insert Figure 4 >>

**4 Towards global soil data interoperability**

[revised manuscript text omitted]

**Appendix A: Naming conventions and descriptions of variables provided in the "WOSIS July 2016" snapshot**

| Code [a] | Attribute | Unit | Profiles | Layers | Description |
|---|---|---|---|---|---|
| BDFI | Bulk density fine earth | $kg\ dm^{-3}$ | 20 727 | 105 848 | Bulk density of the fine earth fraction < 2 mm ($kg/dm^3$) |
| BDWS | Bulk density whole soil | $kg\ dm^{-3}$ | 25 909 | 153 568 | Bulk density of the whole soil including coarse fragments ($kg/dm^3$) |
| TCEQ | Calcium carbonate equivalent total | $g\ kg^{-1}$ | 27 809 | 115 448 | The content of carbonate in a liming material or calcareous soil calculated as if all of the carbonate is in the form of $CaCO_3$ (g/kg in the fine earth fraction < 2 mm); also known as inorganic carbon |
| CECX | Cation exchange capacity (CEC) | $cmol_{(c)}\ kg^{-1}$ | 48 461 | 273 346 | Capacity of the fine earth fraction < 2 mm to hold exchangeable cations, estimated by buffering the soil at specified pH (e.g. pH7 or pH8; $cmol_c/kg$) |
| CLAY | Clay total | $g\ kg^{-1}$ | 80 082 | 408 452 | Gravimetric content of < 0.002 mm soil material in the fine earth fraction < 2 mm (g/100g) |
| CFGR | Coarse fragments gravimetric total | $10^{-2}\ g\ g^{-1}$ | 27 050 | 159 206 | Gravimetric content of coarse fragments > 2 mm in the whole soil (g/100g) |
| CFVO | Coarse fragments volumetric total | $10^{-2}\ cm^3\ cm^{-3}$ | 37 280 | 198 534 | Volumetric content of the coarse fragments > 2 mm in the whole soil ($cm^3/100cm^3$) |
| ECEC | Effective cation exchange capacity (ECEC) | $cmol_{(c)}\ kg^{-1}$ | 23 189 | 102 665 | Capacity of the fine earth fraction < 2 mm to hold exchangeable cations at the pH of the soil (ECEC, $cmol_c/kg$). Conventionally approximated by summation of exchangeable bases ($Ca^{2+}$, $Mg^{2+}$, $K^+$, and $Na^+$) plus 1 N KCl exchangeable acidity ($Al^{3+}$ and $H^+$) in acidic soils |
| ELCO | Electrical conductivity | $dS\ m^{-1}$ | 28 266 | 120 039 | Ability of a 1:x soil water extract to conduct electrical current ($EC_x$, dS/m); $EC_e$ refers to values measured in a saturated soil extract |
| ORGC | Organic carbon | $g\ kg^{-1}$ | 64 118 | 315 362 | Gravimetric content of organic carbon in the fine earth fraction |
| PHCA | pH $CaCl_2$ | unitless | 39 074 | 237 756 | A measure of the acidity or alkalinity in soils, defined as the negative logarithm (base 10) of the activity of hydronium ions ($H^+$) in a $CaCl_2$ solution, as specified in the analytical method descriptions (dimensionless) |
| PHAQ | pH $H_2O$ | unitless | 79 118 | 407 226 | A measure of the acidity or alkalinity in soils, defined as the negative logarithm (base 10) of the activity of hydronium ions ($H^+$) in water (dimensionless) |
| PHKC | pH KCl | unitless | 19 064 | 88 127 | A measure of the acidity or alkalinity in soils, defined as the negative logarithm (base 10) of the activity of hydronium ions ($H^+$) in a KCl solution, as specified in the analytical method descriptions (dimensionless) |
| PHNF | pH NaF | unitless | 4866 | 24 917 | A measure of the acidity or alkalinity in soils, defined as the negative logarithm (base 10) of the activity of hydronium ions ($H^+$) in a NaF solution, as specified in the analytical method descriptions (dimensionless) |
| SAND | Sand total | $10^{-2}\ g\ g^{-1}$ | 78 402 | 398 573 | Larger than Y mm fraction of the < 2 mm soil material (g/100g); esd (equivalent spherical diameter), Y as specified in the analytical method descriptions |
| SILT | Silt total | $10^{-2}\ g\ g^{-1}$ | 79 331 | 406 502 | 0.002 mm to Y mm fraction of the < 2 mm soil material (g/100g); esd (equivalent spherical diameter), X respectively Y as specified in the analytical method descriptions |

| Code[a] | Attribute | Unit | Profiles | Layers | Description |
|---|---|---|---|---|---|
| CFAO | Soil classification FAO | unitless | 24 894 | 24 894 | Classification of the soil profile according to specified edition (year) of the FAO-Unesco Legend, up to soil unit level when available |
| CSTX | Soil classification Soil taxonomy | unitless | 21 614 | 21 614 | Classification of the soil profile according to specified edition (year) of USDA Soil Taxonomy, up to subgroup level when available |
| CWRB | Soil classification WRB | unitless | 24 628 | 24 628 | Classification of the soil profile according to specified edition (year) of the World Reference Base for Soil Resources (WRB), up to qualifier level when available |
| TOTC | Total carbon | $g\ kg^{-1}$ | 14 094 | 70 687 | Gravimetric content of organic carbon and inorganic carbon in the fine earth fraction < 2 mm (g/kg) |
| WRGR | Water retention gravimetric | $10^{-2}\ g\ g^{-1}$ | 28 701 | 173 972 | Soil moisture content by weight, at the tension specified in the analytical method descriptions (g/100g) |
| WRVO | Water retention volumetric | $10^{-2}\ cm^3\ cm^{-3}$ | 17 124 | 82 695 | Soil moisture content by volume, at the tension specified in the analytical method descriptions ($cm^3/100cm^3$) |

[a]   A full complement of all these attributes is generally not available for many profiles (see text).

**Appendix B: Structure of the "July 2016" WoSIS snapshot**

This Appendix describes the structure of the data files presented in the "July 2016" WoSIS snapshot:

- *wosis_201607_attributes.txt*
- *wosis_201607_profiles.txt*, and
- *wosis_201607_layers.txt*.

The first file lists the four letter code for each attribute, a short explanation, and the units of measurement, respectively (Appendix A). This file also gives the number of profiles and layers in the present snapshot.

The second file lists the unique profile ID (i.e. primary key), country name and ISO country code, geometric accuracy, latitude and longitude (WGS 1984) as well as information on the soil classification system and edition. Depending on the soil classification system used, the number of fields will vary. For example, for US Soil Taxonomy, coded here as "cstx", these are: order, suborder, great group and subgroup as indicated in the column headings.

The third, largest file, lists all the soil properties by layer and profile to enhance user-friendliness. It starts with:

| | |
|---|---|
| profile_id | identifier for profile, links to file wosis_201607_profiles. |
| profile_layer_id | unique identifier for layer for given profile (primary key) |
| top | upper depth of layer (or horizon) |
| bottom | lower depth of layer |

Subsequently, the following items are listed sequentially per attribute ("xxxx") as defined under "code" in file *wosis_201607_attributes.txt*:

| | |
|---|---|
| xxxx_value | array listing all values for soil property "xxxx" for the given layer; thus, more than one observation can be reported when available, for example 3 values for ORGC: {1:0.55, 2:1.01, 3:0.85} |
| xxxx_value_avg | average, for above (use this value for "routine" modelling) |
| xxxx_method | array listing the method descriptions for each value |
| xxxx_date | array listing the date of observation for each value |
| xxxx_dataset_id | abbreviation for source data set (e.g. WD-ISIS), |
| xxxx_profile_code | code for given profile |
| xxxx_licenset | licence for given data (CC-BY-NC or CC-BY) |

(... )                    as above, but for the next attribute

All fields in the above files are tab-delimited, while double quotation marks serve as text delimiters; file coding is according to the UTF-8 unicode transformation format. As such, the files can be easily imported into an SQL database or statistical software such as R, after which they may be joined using the unique profile_id.

**Appendix C: Number of profiles by country and continent.**

| Continent | Country_name | ISO code | N of profiles | Area (km$^2$) | Profile density (per 1000 km$^2$) |
|---|---|---|---|---|---|
| Africa | Algeria | DZ | 4 | 2 308 647 | 0.002 |
| | Angola | AO | 1035 | 1 246 690 | 0.830 |
| | Benin | BJ | 738 | 115 247 | 6.404 |
| | Botswana | BW | 898 | 578 247 | 1.553 |
| | Burkina Faso | BF | 887 | 273 281 | 3.246 |
| | Burundi | BI | 36 | 26 857 | 1.340 |
| | Cameroon | CM | 455 | 465 363 | 0.978 |
| | Central African Republic | CF | 87 | 619 591 | 0.140 |
| | Chad | TD | 5 | 1 265 392 | 0.004 |
| | Congo | CG | 70 | 340 599 | 0.206 |
| | Côte d'Ivoire | CI | 254 | 321 762 | 0.789 |
| | Dem. Rep. of the Congo | CD | 374 | 2 329 162 | 0.161 |
| | Egypt | EG | 22 | 98 2161 | 0.022 |
| | Ethiopia | ET | 1583 | 1 129 314 | 1.402 |
| | Gabon | GA | 46 | 264 022 | 0.174 |
| | Ghana | GH | 163 | 238 842 | 0.682 |
| | Guinea | GN | 62 | 243 023 | 0.255 |
| | Guinea-Bissau | GW | 18 | 30 740 | 0.586 |
| | Kenya | KE | 504 | 582 342 | 0.865 |
| | Lesotho | LS | 33 | 30 453 | 1.084 |
| | Liberia | LR | 48 | 96 103 | 0.499 |
| | Libya | LY | 14 | 1 620 583 | 0.009 |
| | Madagascar | MG | 52 | 588 834 | 0.088 |
| | Malawi | MW | 2985 | 118 715 | 25.144 |
| | Mali | ML | 756 | 1 251 471 | 0.604 |
| | Mauritania | MR | 11 | 1 038 527 | 0.011 |
| | Morocco | MA | 27 | 414 030 | 0.065 |
| | Mozambique | MZ | 275 | 787 305 | 0.349 |
| | Namibia | NA | 62 | 823 989 | 0.075 |
| | Niger | NE | 488 | 1 182 602 | 0.413 |
| | Nigeria | NG | 1203 | 908 978 | 1.323 |
| | Rwanda | RW | 92 | 25 388 | 3.624 |
| | Senegal | SN | 311 | 196 200 | 1.585 |
| | Sierra Leone | SL | 12 | 72 281 | 0.166 |
| | Somalia | SO | 245 | 632 562 | 0.387 |
| | South Africa | ZA | 649 | 1 220 127 | 0.532 |
| | Sudan | SD | 116 | 1 843 196 | 0.063 |
| | Swaziland | SZ | 14 | 17 290 | 0.810 |
| | Togo | TG | 9 | 56 767 | 0.159 |
| | Tunisia | TN | 60 | 155 148 | 0.387 |
| | Uganda | UG | 12 | 241 495 | 0.050 |

| Continent | Country_name | ISO code | N of profiles | Area (km$^2$) | Profile density (per 1000 km$^2$) |
|---|---|---|---|---|---|
| | United Republic of Tanzania | TZ | 1647 | 939 588 | 1.753 |
| | Zambia | ZM | 472 | 751 063 | 0.628 |
| | Zimbabwe | ZW | 319 | 390 648 | 0.817 |
| Antarctica | Antarctica | AQ | 9 | 12 537 967 | 0.001 |
| Asia | Afghanistan | AF | 19 | 641 827 | 0.030 |
| | Armenia | AM | 6 | 29 624 | 0.203 |
| | Azerbaijan | AZ | 4 | 164 780 | 0.024 |
| | Bahrain | BH | 2 | 673 | 2.970 |
| | Bangladesh | BD | 16 | 139 825 | 0.114 |
| | Bhutan | BT | 80 | 37 674 | 2.123 |
| | China | CN | 1490 | 9 345 214 | 0.159 |
| | Georgia | GE | 9 | 69 785 | 0.129 |
| | India | IN | 139 | 2 961 118 | 0.047 |
| | Indonesia | ID | 108 | 1 888 620 | 0.057 |
| | Iran (Islamic Republic of) | IR | 2 | 1 677 319 | 0.001 |
| | Iraq | IQ | 14 | 435 864 | 0.032 |
| | Israel | IL | 16 | 20 720 | 0.772 |
| | Japan | JP | 39 | 373 651 | 0.104 |
| | Jordan | JO | 40 | 89 063 | 0.449 |
| | Lebanon | LB | 6 | 10 136 | 0.592 |
| | Malaysia | MY | 46 | 329 775 | 0.139 |
| | Mongolia | MN | 7 | 1 564 529 | 0.004 |
| | Nepal | NP | 141 | 147 437 | 0.956 |
| | Oman | OM | 9 | 308 335 | 0.029 |
| | Pakistan | PK | 43 | 788 439 | 0.055 |
| | Philippines | PH | 68 | 296 031 | 0.230 |
| | Republic of Korea | KR | 17 | 99 124 | 0.172 |
| | Sri Lanka | LK | 13 | 66 173 | 0.196 |
| | Syrian Arab Republic | SY | 66 | 188 128 | 0.351 |
| | Taiwan | TW | 33 | 36 127 | 0.913 |
| | Tajikistan | TJ | 5 | 142 004 | 0.035 |
| | Thailand | TH | 285 | 515 417 | 0.553 |
| | Turkey | TR | 68 | 781 229 | 0.087 |
| | United Arab Emirates | AE | 6 | 71 079 | 0.084 |
| | Uzbekistan | UZ | 8 | 449 620 | 0.018 |
| | Yemen | YE | 284 | 453 596 | 0.626 |
| Europe | Albania | AL | 63 | 28 682 | 2.197 |
| | Belarus | BY | 94 | 207 581 | 0.453 |
| | Belgium | BE | 190 | 30 669 | 6.195 |
| | Bulgaria | BG | 45 | 111 300 | 0.404 |
| | Czech Republic | CZ | 38 | 78 845 | 0.482 |

| Continent | Country_name | ISO code | N of profiles | Area (km$^2$) | Profile density (per 1000 km$^2$) |
|---|---|---|---|---|---|
| | Denmark | DK | 20 | 44 458 | 0.450 |
| | Estonia | EE | 123 | 45 441 | 2.707 |
| | Finland | FI | 24 | 336 892 | 0.071 |
| | France | FR | 53 | 548 785 | 0.097 |
| | Germany | DE | 51 | 357 227 | 0.143 |
| | Greece | GR | 11 | 132 549 | 0.083 |
| | Hungary | HU | 61 | 93 119 | 0.655 |
| | Iceland | IS | 11 | 102 566 | 0.107 |
| | Ireland | IE | 36 | 69 809 | 0.516 |
| | Italy | IT | 86 | 301 651 | 0.285 |
| | Latvia | LV | 10 | 64 563 | 0.155 |
| | Lithuania | LT | 18 | 64 943 | 0.277 |
| | Luxembourg | LU | 128 | 2621 | 48.842 |
| | Netherlands | NL | 192 | 35 203 | 5.454 |
| | Norway | NO | 10 | 324 257 | 0.031 |
| | Poland | PL | 128 | 311 961 | 0.410 |
| | Portugal | PT | 35 | 91 876 | 0.381 |
| | Republic of Moldova | MD | 32 | 33 798 | 0.947 |
| | Romania | RO | 43 | 238 118 | 0.181 |
| | Russian Federation | RU | 156 | 16 998 830 | 0.009 |
| | Slovakia | SK | 40 | 49 072 | 0.815 |
| | Spain | ES | 42 | 505 752 | 0.083 |
| | Sweden | SE | 26 | 449 212 | 0.058 |
| | Switzerland | CH | 10 | 41 257 | 0.242 |
| | Ukraine | UA | 79 | 600 526 | 0.132 |
| | United Kingdom | GB | 53 | 244 308 | 0.217 |
| North America | Barbados | BB | 3 | 433 | 6.928 |
| | Belize | BZ | 21 | 21 764 | 0.965 |
| | Canada | CA | 148 | 9 875 646 | 0.015 |
| | Costa Rica | CR | 55 | 51 042 | 1.078 |
| | Cuba | CU | 52 | 110 863 | 0.469 |
| | Dominican Republic | DO | 1 | 48 099 | 0.021 |
| | El Salvador | SV | 9 | 20 732 | 0.434 |
| | Greenland | GL | 5 | 2 165159 | 0.002 |
| | Guatemala | GT | 22 | 109 062 | 0.202 |
| | Honduras | HN | 11 | 112 124 | 0.098 |
| | Jamaica | JM | 76 | 10 965 | 6.931 |
| | Mexico | MX | 12 223 | 1 949 527 | 6.270 |
| | Netherlands Antilles | AN | 4 | 790 | 5.066 |
| | Nicaragua | NI | 26 | 128 376 | 0.203 |
| | Panama | PA | 25 | 74 850 | 0.334 |
| | Puerto Rico | PR | 30 | 8937 | 3.357 |
| | Trinidad and Tobago | TT | 2 | 5144 | 0.389 |

| Continent | Country_name | ISO code | N of profiles | Area (km$^2$) | Profile density (per 1000 km$^2$) |
|---|---|---|---|---|---|
| | United States of America | US | 50 361 | 9 315 946 | 5.406 |
| | United States Virgin Islands | VI | 3 | 352 | 8.514 |
| Oceania | Australia | AU | 142 | 7 687 634 | 0.018 |
| | Cook Islands | CK | 1 | 241 | 4.142 |
| | Fiji | FJ | 9 | 18 293 | 0.492 |
| | Micronesia (Feder. States of) | FM | 14 | 740 | 18.917 |
| | New Zealand | NZ | 20 | 270 415 | 0.074 |
| | Papua New Guinea | PG | 31 | 462 230 | 0.067 |
| | Samoa | WS | 17 | 2835 | 5.996 |
| | Solomon Islands | SB | 1 | 28 264 | 0.035 |
| South America | Argentina | AR | 238 | 2 780 175 | 0.086 |
| | Bolivia (Plurinational State of) | BO | 77 | 1 084 491 | 0.071 |
| | Brazil | BR | 7842 | 8 485 946 | 0.924 |
| | Chile | CL | 45 | 753 355 | 0.060 |
| | Colombia | CO | 166 | 1 137 939 | 0.146 |
| | Ecuador | EC | 77 | 256 249 | 0.300 |
| | French Guiana | GF | 7 | 83 295 | 0.084 |
| | Guyana | GY | 43 | 211 722 | 0.203 |
| | Peru | PE | 147 | 1 290 640 | 0.114 |
| | Suriname | SR | 27 | 145 100 | 0.186 |
| | Uruguay | UY | 131 | 177 811 | 0.737 |
| | Venezuela (Bolivarian Rep. of) | VE | 170 | 912 025 | 0.186 |
| World | World (total) | WD | 94 441 | 137 770 610 | 0.685 |

Note: Country names and areas are based on the Global Administrative Layers (GAUL) database, see:

http://www.fao.org/geonetwork/srv/en/metadata.show?id=12691.

**Acknowledgements**

The development of WoSIS has been made possible thanks to the contributions and shared knowledge of a steadily growing number of data providers, including soil survey organisations, research institutes and individual experts, whose contributions are gratefully acknowledged. A detailed list of data providers is available at http://www.isric.org/content/wosis-cooperating-institutions-and-experts for details.

**Figure captions**

**Figure 1**. General procedure for processing data in WoSIS

**Figure 2.** Flagging of repeated profiles between the AfSP, ISIS, WISE and SOTER databases (see text for explanation of abbreviations)

**F**igure **3.** Serving consistent soil layers from WoSIS to the user community through ISRIC's evolving spatial data infrastructure

**Figure 4.** Location of soil profiles provided in the "July 2016" snapshot of WoSIS.
(See Appendix C for the number of profiles by country)

[Figure]

**Figure 1.** General procedure for processing data in WoSIS

[Figure]

**Figure 2**. Flagging of repeated profiles between the AfSP, ISIS, WISE and SOTER databases (see text for explanation of abbreviations)

[Figure]

**Figure 3.** Serving consistent soil layers from WoSIS to the user community through ISRIC's evolving spatial data
infrastructure

[Figure]

**Figure 4**. Location of soil profiles provided in the "July 2016" snapshot of WoSIS.
(See Appendix C for the number of profiles by country)